METHODS AND RESOURCES

# Extent and context dependence of pleiotropy revealed by high-throughput single-cell phenotyping

Kerry A. Geiler-Samerotte[1,2]*, Shuang Li[1,3], Charalampos Lazaris[1,4], Austin Taylor[1], Naomi Ziv[1,5], Chelsea Ramjeawan[1], Annalise B. Paaby[6], Mark L. Siegal[1]

1 Center for Genomics and Systems Biology, Department of Biology, New York University, New York, New York, United States of America, 2 Center for Mechanisms of Evolution, Biodesign Institutes, School of Life Sciences, Arizona State University, Tempe, Arizona, United States of America, 3 Department of Pharmacology, University of North Carolina at Chapel Hill, Chapel Hill, North Carolina, United States of America, 4 Whitehead Institute for Biomedical Research, Cambridge, Massachusetts, United States of America, 5 Department of Microbiology and Immunology, University of California, San Francisco, California, United States of America, 6 School of Biological Sciences, Georgia Institute of Technology, Atlanta, Georgia, United States of America

* kerry.samerotte@asu.edu

**Data Availability Statement:** All data are available at open science framework DOI 10.17605/OSF.IO/B7NY5.

## Abstract

Pleiotropy—when a single mutation affects multiple traits—is a controversial topic with far-reaching implications. Pleiotropy plays a central role in debates about how complex traits evolve and whether biological systems are modular or are organized such that every gene has the potential to affect many traits. Pleiotropy is also critical to initiatives in evolutionary medicine that seek to trap infectious microbes or tumors by selecting for mutations that encourage growth in some conditions at the expense of others. Research in these fields, and others, would benefit from understanding the extent to which pleiotropy reflects inherent relationships among phenotypes that correlate no matter the perturbation (vertical pleiotropy). Alternatively, pleiotropy may result from genetic changes that impose correlations between otherwise independent traits (horizontal pleiotropy). We distinguish these possibilities by using clonal populations of yeast cells to quantify the inherent relationships between single-cell morphological features. Then, we demonstrate how often these relationships underlie vertical pleiotropy and how often these relationships are modified by genetic variants (quantitative trait loci [QTL]) acting via horizontal pleiotropy. Our comprehensive screen measures thousands of pairwise trait correlations across hundreds of thousands of yeast cells and reveals ample evidence of both vertical and horizontal pleiotropy. Additionally, we observe that the correlations between traits can change with the environment, genetic background, and cell-cycle position. These changing dependencies suggest a nuanced view of pleiotropy: biological systems demonstrate limited pleiotropy in any given context, but across contexts (e.g., across diverse environments and genetic backgrounds) each genetic change has the potential to influence a larger number of traits. Our method suggests that exploiting pleiotropy for applications in evolutionary medicine would benefit from focusing on traits with correlations that are less dependent on context.

**Funding:** This work was supported by National Institutes of Health grant R35GM118170 (to MLS), National Institutes of Health fellowship F32GM103166 (to KAG-S), National Institutes of Health grant R35GM133674 (to KAG-S), a New York University Graduate School of Arts and Science Dean's Dissertation Fellowship (to SL), and National Institutes of Health grant R35GM119744 (to ABP). The funders had no role in study design, data collection and analysis, decision to publish, or preparation of the manuscript.

**Competing interests:** The authors have declared that no competing interests exist.

**Abbreviations:** GdA, geldanamycin; GFP, green fluorescent protein; IQR, interquartile range; LDL, low-density lipoprotein; MA, mutation-accumulation; PCA, principal component analysis; PKU, phenylketonuria; QTL, quantitative trait loci.

## Introduction

Pleiotropy exists when a single mutation affects multiple traits [1,2]. Often, pleiotropy is defined instead as a single gene contributing to multiple traits, although what is implied is the original definition—that a single change at the genetic level can have multiple consequences at the phenotypic level [2]. As our ability to survey the influence of genotype on phenotype improves, examples of pleiotropy are growing [3–8]. For example, individual genetic variants have been associated with seemingly disparate immune, neurological, and digestive symptoms in humans and mice [9,10]. Genes affecting rates of cell division across diverse environments and drug treatments have been identified in microbes and cancers [11,12]. A view emerging from genome-wide association studies is that variation in complex traits is "omnigenic" in the sense that many loci indirectly contribute to variation in many traits [13,14].

However, the extent of pleiotropy remains a major topic of debate because, despite its apparent prevalence, pleiotropy is thought to be evolutionarily disadvantageous. The more traits a mutation affects, the more likely it is that the mutation will have a negative impact on at least one. Pervasive pleiotropy should therefore constrain evolution [15], exacting what is known as a cost of complexity or cost of pleiotropy [11,16–19]. This cost may bias which mutations underlie adaptation, for example, toward less-pleiotropic *cis*-regulatory changes over more-pleiotropic changes in *trans*-acting factors [20,21], or toward changes to proteins that participate in relatively few biological processes [22,23]. Over long periods, the cost of pleiotropy may influence the organization of biological systems, favoring a modular structure in which genetic changes influencing one group of traits have minimal impact system-wide [24–29].

At stake in the ongoing debate about the extent of pleiotropy [30–33] are some of modern biology's prime objectives, including the prediction of complex phenotypes from genotype data [18,34,35] and the prediction of how organisms will adapt to environmental change [36,37]. These predictions are more challenging if genetic changes influence a large number of traits with complex interdependencies. Nonetheless, understanding how a given mutation influences multiple traits could be powerful, allowing prediction of some phenotypic responses given others [38,39]. Indeed, recent strategies in medicine called evolutionary traps aim to exploit pleiotropy, for example, by finding genetic changes that provide resistance to one treatment while promoting susceptibility to another [40–42].

The lack of consensus about the extent of pleiotropy in natural systems is, in part, due to poorly defined expectations for how to test for it experimentally. One key issue is that defining a phenotype is not trivial [43,44]. Consider a variant in the *apolipoprotein B* gene that increases low-density lipoprotein (LDL) cholesterol levels as well as the risk of heart disease. Elevated LDL promotes heart disease [45], so are these two phenotypes or one? Alternatively, consider a mutation in the *phenylalanine hydroxylase* gene that affects nervous system function and skin pigmentation. These dissimilar effects, both symptoms of untreated phenylketonuria (PKU), originate from the same problem: a deficiency in converting phenylalanine to tyrosine [46]. Is it appropriate to call mutations that have this single metabolic effect pleiotropic? Likewise, shall one call pleiotropic a mutation that makes tomatoes both ripen uniformly and taste bad, when the effect of the mutation is to reduce the function of a transcription factor that promotes chloroplast development, which in turn necessarily affects both coloration and sugar accumulation [47]?

The LDL, PKU, and tomato cases are examples of vertical pleiotropy, i.e., pleiotropy that results when one phenotype influences another or both are influenced by a shared factor [5,43]. The alternative to vertical pleiotropy is horizontal pleiotropy, in which genetic differences induce correlations between otherwise independent phenotypes. It might be tempting to

discard vertical pleiotropy as less "genuine" [48] or less important than horizontal pleiotropy, but that would be a mistake because vertical pleiotropy reveals important information about the underlying biological systems that produce the phenotypes in question. Consider the value in identifying yet-unknown factors in heart disease by finding traits that correlate with it, or in understanding where in a system an intervention is prone to produce undesirable side effects. Consider also that the extent and nature of vertical pleiotropy speak directly to the question of modularity: modularity is implied if vertical pleiotropy either is rare or manifests as small groups of correlated traits that are isolated from other such groups. If there is modularity, then there can be horizontal pleiotropy, when particular genetic variants make links between previously unconnected modules.

The above considerations suggest that a unified analysis that distinguishes and compares horizontal and vertical pleiotropy is needed to make sense of the organization and evolution of biological systems. However, existing methods of distinguishing horizontal and vertical pleiotropy are problematic because judgments must be made about which traits are independent from one another. Such judgments differ between researchers and over time. Indeed, the tomato example can be viewed as a case of horizontal pleiotropy transitioning recently to vertical pleiotropy as knowledge of the underlying system advanced.

In this study, we propose and apply an empirical and analytical approach to measuring pleiotropy that relies far less on subjective notions of what constitutes an independent phenotype. The key principle is that the distinction between vertical and horizontal pleiotropy lies in whether traits are correlated in the absence of genetic variation [43]. For vertical pleiotropy, the answer is yes: because one trait influences the other or the two share an influence, nongenetic perturbations that alter one phenotype are expected to alter the other. For horizontal pleiotropy, the answer is no: genetic variation causes the trait correlation. In this study, we determined how traits correlate in the absence of genetic variation by measuring single-cell traits in clonal populations of cells.

We used high-throughput morphometric analysis [49–53] of hundreds of thousands of single cells of the budding yeast *Saccharomyces cerevisiae* to measure how dozens of cell-morphology traits (thousands of pairs of traits) covary within clonal populations and between such populations representing different genotypes. Within-genotype correlations report on vertical pleiotropy, whereas between-genotype correlations report on horizontal pleiotropy to the extent that they exceed the corresponding within-genotype correlations. For one set of genotypes, we used 374 progeny of a cross of two natural isolates [54], which enabled not only the estimation of vertical and horizontal pleiotropy but also the identification of quantitative trait loci (QTL) with pleiotropic effects. For another set of genotypes, we used a collection of mutation-accumulation (MA) lines, each of which contains a small number of unique spontaneous mutations [55,56], which enabled a more direct test of the ability of mutations to alter trait correlations.

The traits we study—morphological features of single cells—represent important fitness-related traits [51,57,58] that contribute to processes such as cell division and tissue invasion (e.g., cancer metastasis [59]). Cell-morphological features may correlate across cells for a variety of vertical or horizontal reasons. Vertical reasons include (1) inherent geometric constraints (e.g., on cell circumference and area); (2) constraints imposed by gene-regulatory networks (e.g., if the genes influencing a group of traits are all under control of the same transcription factor); and (3) constraints induced by developmental processes (e.g., as a yeast cell divides or "buds," many morphological features are affected). Horizontal pleiotropy might be evident because genetic variants each affecting two or more traits (that are otherwise weakly correlated) are segregating in the progeny of the cross between two natural isolates. Alternatively, horizontal pleiotropy might be evident because a particular allele strengthens the trait

correlation so that genetic variation affecting one trait is more likely to affect another when that allele is present. These alternatives can be distinguished by examining trait correlations in two subsets of progeny strains defined by which natural isolate's allele they possess at a QTL of interest.

In addition to genetic variation, nongenetic variation may also alter the correlations between traits. We rely on nongenetic heterogeneity within clonal populations to serve as perturbations that reveal inherent trait correlations. However, the correlations themselves might be heterogeneous within these populations. For example, the dependencies between morphological features may change as cells divide. To control for this possibility, we performed our trait mapping and subsequent analysis after binning cells into three stages (unbudded, small-budded, and large-budded cells). We further examined whether trait correlations change across the cell cycle by using a machine-learning approach to more finely bin the imaged cells into 48 stages of division.

Collectively, the results we present here demonstrate that both types of pleiotropy, vertical and horizontal, are prevalent for single-cell morphological traits, suggesting that biological systems occupy a middle ground between extreme modularity and extreme interconnectedness. Perhaps more surprisingly, we find that trait correlations are often context dependent and can be altered by mutations as well as cell-cycle state and drug treatments. The dynamic nature of trait correlations encourages caution when attempting to quantify and interpret the extent of pleiotropy in nature or when making predictions about correlated phenotypic responses to the same selection pressure, as is done when crafting evolutionary traps. However, applying our approach may suggest which trait correlations are less context dependent and therefore more useful in setting such traps.

## Results

### QTLs with pleiotropic effects influence yeast single-cell morphology

To detect genes with pleiotropic effects on cell morphology, we measured 167 single-cell morphological features (e.g., cell size, bud size, bud angle, distance from nucleus to bud neck; S1 Table) in 374 yeast strains that were generated in a previous study from a mating between two wild yeast isolates [54,60]. These wild isolates, one obtained from soil near an oak tree, the other from a wine barrel, differ by 0.006 SNPs per site [61] and have many heritable differences in single-cell morphology [62]. For example, we find that yeast cells from the wine strain, on average, are smaller, are rounder, and have larger nuclei during budding than yeast cells from the oak strain (S1 Fig).

To measure their morphologies, we harvested exponentially growing cells from three replicate cultures of each of these 374 recombinant strains and imaged, on average, 800 fixed, stained cells per strain using high-throughput microscopy in a 96-well plate format (S2 Fig). We used control strains present on each plate to correct for plate-to-plate variation and quantified morphological features using CalMorph software [53], which divides cells into three categories based on their progression through the cell cycle (i.e., unbudded, small-budded, and large-budded cells) and measures phenotypes specific to each category.

A simple way to measure pleiotropy would be to identify QTL that contribute to variation in these phenotypes and then to count the number of phenotypes to which each QTL contributes. However, such a measure is sensitive to the statistical thresholds that are used and therefore risks yielding false inferences about trait modularity. Using a liberal threshold would cause false-positive cases of pleiotropy (less apparent modularity), whereas a conservative threshold would cause failures to detect pleiotropic QTL when they exist (more apparent modularity). Such a measure also assumes the counted traits are somehow independent except for

correlations induced by genetic variants. Statistically independent traits could be constructed (and then counted) as principal components of the original traits, but the concern about too-liberal or too-conservative QTL-detection thresholds would remain. Moreover, as we explore extensively below, trait correlations are hierarchical (differing within and between genotypes and conditions), making application of principal components analysis problematic. For these reasons, we do not focus on counting the number of phenotypes influenced by a given locus. Still, to begin to dissect vertical and horizontal pleiotropy, we must start with candidate examples of pleiotropic loci.

To detect QTL, we used 225 markers spread throughout the genome [54] and Haley-Knott regression implemented in the R package R/qtl [63,64]. We used a standard permutation-based method to estimate statistical significance [63–65], with permutations performed separately for each trait such that the per-trait probability of detecting a false positive is 0.05. With this cutoff, we identified 41 QTL that contribute to variation in 155 of the surveyed morphological features (Fig 1A). This approach does not correct for the testing of multiple traits. When we do so using a false-discovery rate set to 5%, results do not change qualitatively. Indeed, the majority of QTL-trait associations that are eliminated using this more-stringent cutoff involve QTL that are detected regardless of this correction (Fig 1A; 80% of red points are present at QTL that also possess black points). This observation suggests many of the associations detected with the less-stringent cutoff are not spurious. We therefore present subsequent analyses using the QTL-trait associations based on the less-stringent cutoff, but we also report analyses using the reduced set of associations based on the more-stringent cutoff to establish that qualitative conclusions did not change.

Most of the QTL we detect are pleiotropic, meaning each contributes to variation in more than one morphological feature (Fig 1A; 36/41 QTL influence multiple traits, 20/26 after correcting for testing multiple traits). The median number of traits to which each QTL contributes is 5 (5.5 after correction). This finding provides some support for the idea that biological systems demonstrate limited pleiotropy, in that the median number of traits affected per QTL is low (5/167). This median number of traits is similar to that found in previous high-dimensional QTL screens [17,31] and in analyses of gene knockouts in yeast and mouse [11]. Further evidence that biological systems demonstrate limited pleiotropy, and thus a modular organization, comes from previous studies of this mapping family that show these same QTL do not contribute to variation in sporulation efficiency [60].

However, as noted above, conclusions drawn about modularity from studies that count traits are subject to criticism: some QTL influences may be too small to detect, even with less-stringent significance thresholds, creating the appearance of modularity even if every QTL influences every trait to at least some small degree. Further, we detect some QTL that influence large numbers of traits, up to 73 (68 after correction). Although such QTL mitigate, to some extent, concerns about detection power, they highlight other potential problems: some QTL might contain multiple genetic differences that impact different traits, and some morphological features might be inherently correlated and therefore should not be counted as independent traits. Next, we focus on the first of these problems by asking whether these QTL represent single genes that contribute to phenotypic variation in many morphological features. We return to the issue of trait independence after that.

## Single genes with pleiotropic effects influence yeast single-cell morphology

When a QTL affects multiple traits, it might not mean that variation in a single gene is contributing to variation in these traits but instead that linked genes are contributing to variation in distinct, individual traits. We partitioned genotype-phenotype associations on the same

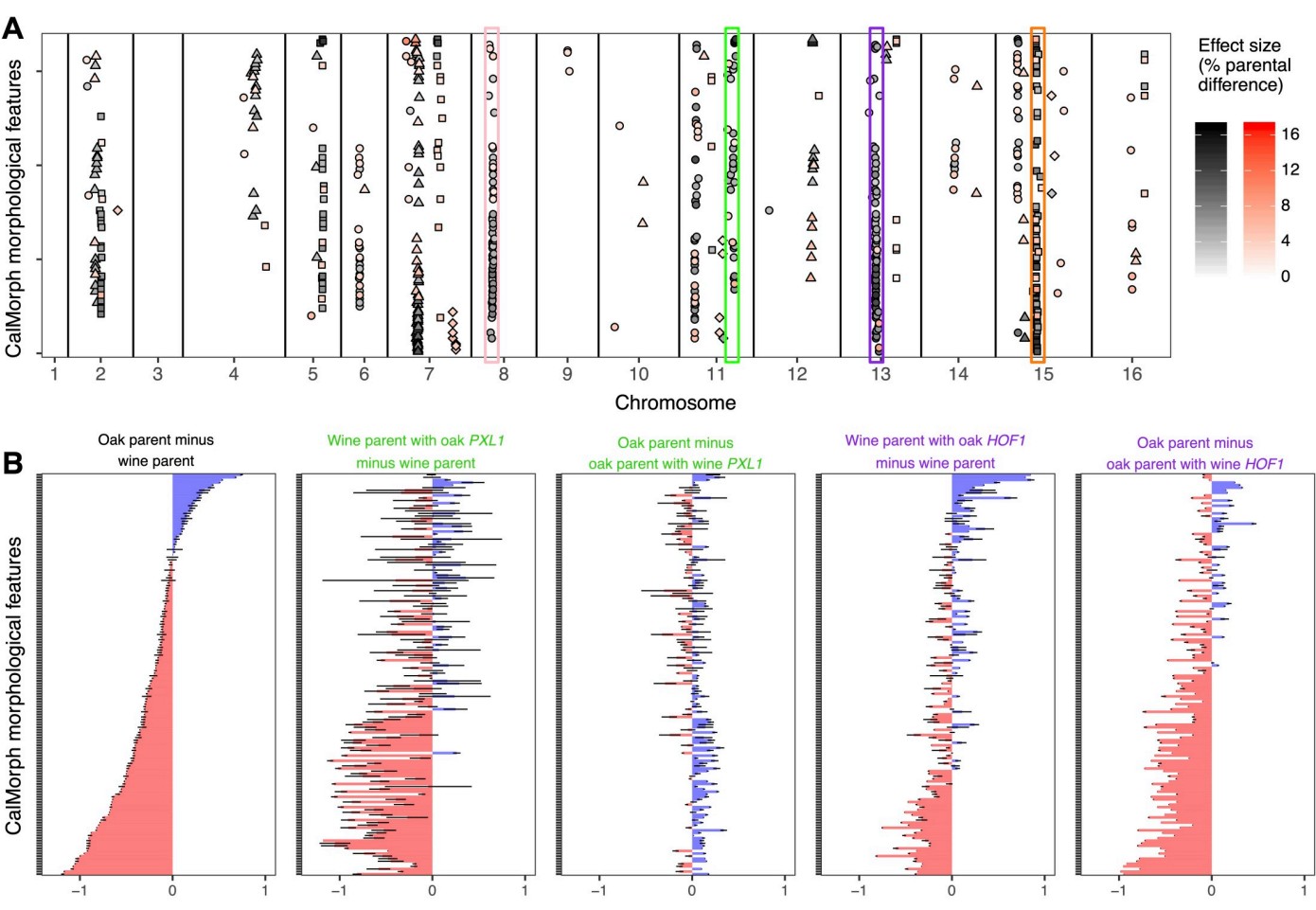

**Fig 1. Pleiotropic QTL influence yeast single-cell morphology.** The vertical axes in all plots represent the 155 CalMorph morphological traits for which we detect QTL. These traits are sorted, from top to bottom, based on the difference between the oak and wine parental strains. (A) Of 41 QTL that contribute to variation in single-cell morphology, 36 contribute to variation in multiple features. The horizontal axis indicates the chromosomal location of each QTL (in cM). Differently shaped points indicate separate QTL that are more than 5 cM apart on the same chromosome. The darkness of a point represents the effect size of a QTL; effect sizes range from 1.3% (lightest points) to 17.5% (darkest points) of the difference between parents. All points (red and black) represent genotype-phenotype associations detected using a per-trait genome-wide type I error rate of 5%. The points highlighted in black are significant after correcting for testing multiple phenotypes using a false-discovery rate of 5%. The QTL highlighted in pink, green, purple, and orange are very pleiotropic, contributing to 57, 30, 73, or 64 morphological features, respectively. (B) Gene-swapping experiments demonstrate that single genes contribute to multiple morphological features. The horizontal axis represents the relative phenotypic differences between the wine and oak parents (leftmost column) or one of these strains versus a derivative strain that differs in a single gene. The relative phenotypic differences between a pair of strains are calculated by scaling each trait to have a mean of 0 and standard deviation of 1 across all cells in both strains, and then subtracting the average value in one strain from that in the other. To control for variation among replicate experiments, this scaling was done independently for each replicate experiment in which both strains were imaged. Error bars represent 95% confidence intervals inferred from the replicate experiments. The two gene replacements shown, *PXL1* and *HOF1*, are respectively located within the QTL highlighted in green and purple in (A). When calculating the difference between strains, we always subtracted the trait values of the strain possessing more wine genes from those of the strain possessing more oak genes, such that the effects of the wine or oak gene replacements appear in the same direction on all plots. Data underlying this figure can be found at https://osf.io/b7ny5/. QTL, quantitative trait loci.

chromosome into separate QTL when they were greater than 5 cM apart, except in genome regions where many genotype-phenotype associations are present and there is no clear break point. A distance of 5 cM corresponds on average to eight protein-coding genes in the yeast genome. The largest QTL we detected spans 17 cM, which is roughly half the window size utilized in a previous study of this same QTL-mapping family [63]. This approach reduces but does not eliminate the possibility that QTL represent the action of linked loci.

For a small number of QTL with high pleiotropy (highlighted in Fig 1A), we sought to test whether the effects on different morphological features were due to the action of a single gene. We performed these tests by swapping the parental versions of candidate genes (i.e., we genetically modified the wine strain to carry the oak version of a given gene, and vice versa). We used the *delitto perfetto* technique to perform these swaps [66], such that the only difference between a parental genome and the swapped genome is the coding sequence of the single candidate gene plus up to 1 kb of flanking sequence. Candidate genes were selected based on descriptions of the single-cell morphologies of their knockout mutants [67] and the presence of at least one nonsynonymous amino acid difference between the wine and oak alleles [62].

When a candidate gene contributes to the morphological differences between the wine and oak parents, we expect yeast strains that differ at only that locus to recapitulate some of the morphological differences between the wine and oak parents. Indeed, this is what we observe for *PXL1*, a candidate for the QTL on chromosome 11, and *HOF1*, a candidate for the QTL on chromosome 13 (Fig 1B; compare each plot on the right to the leftmost plot; see also S2 Table). This influence is most pervasive for *HOF1*; both the oak and the wine alleles have a strong effect on the morphology of the opposite parent, and their effects recapitulate the parental difference to a large extent. The pervasive influence of *HOF1* on various morphological features is consistent with the fact that this gene's product affects actin-cable organization and is involved in both polar cell growth and cytokinesis [68]. The effect of *PXL1* on cell morphology is also apparent across many single-cell features, although only the oak allele has a strong effect that recapitulates the parental difference. We evaluated *RAS1*, a candidate for the QTL on chromosome 15, but initial tests indicated that it did not have a significant impact on most morphological features (S2 Table). We also attempted to swap alleles for a candidate gene corresponding to the QTL on chromosome 8 but were unsuccessful.

A previous screen for QTL influencing single-cell morphology in the progeny of a genetically distinct pair of yeast strains (a different vineyard strain and a laboratory strain) found some of the same pleiotropic QTL that we detect in the wine and oak cross [69] (compare their Table 2 to our S1 Table). In particular, we both find a QTL in the same position on chromosome 15 that influences many morphological features related to nucleus size, shape, and position in the cell (Fig 1A; orange). We also both detect a QTL near base pair 100,000 on chromosome 8 that influences cell size and shape (Fig 1A; pink). In the previous screen, the genetic basis of this QTL was shown to be a single nucleotide change within the *GPA1* gene [69].

The main conclusion from our gene-swapping experiments, which is consistent with the previous cell-morphology QTL study [69] as well as with comprehensive surveys of how gene deletions affect the morphology of a laboratory yeast strain [11,49], is that single genes with pleiotropic effects on cell morphology are readily detected in budding yeast. Moreover, the morphological traits involved were previously shown to influence fitness [51,57,58], which raises the question: why do so many genetic analyses (including ours) detect pleiotropy [5,9–12,14] when other work suggests that pleiotropy exacts a cost [17,18,20,21]?

## Dissecting pleiotropy using clonal populations of cells

One hypothesis to explain pervasive pleiotropy may be that the phenotypes we chose to measure are not independent. Instead, many of these single-cell morphological features may be inherently related such that perturbing one will have unavoidable consequences on another and thus any associated limitation of adaptation will be unavoidable as well. In other words, the hypothesis is that much of the pleiotropy we observe is vertical pleiotropy. A test of this hypothesis is to ask whether traits that are jointly affected by the same QTL are correlated in the absence of genetic differences. Our dataset provides a unique opportunity to perform such

a test because we quantified single-cell traits for, on average, 800 clonal cells per yeast strain (S2 Fig).

We leverage the hierarchical structure and large sample size of our dataset to obtain precise estimates of the correlations that exist within and between strains. Thereby we learn about the underlying relationships between morphological traits, which we use to distinguish vertical from horizontal pleiotropy. Because we are studying clonal families without a complicated pedigree structure, these within- and between-strain correlations are equivalent to the so-called environmental and genetic correlations of quantitative genetics [70]. Here, we use a simple (and fast) method that is appropriate for two-level hierarchical data to partition the total correlation into a pooled within-strain component ($r_W$) and a between-strain component ($r_B$) [71]. One caveat of this correlation-partitioning approach is that $r_B$ is effectively the correlation between strain means, which can bias estimates of genetic covariance [70]. This bias is most pronounced at small sample sizes [70], so our large sample sizes allay concern. Nonetheless, for a subset of traits, we tested whether estimates obtained from correlation partitioning are similar to those obtained from mixed-effect linear models that specify the variance-covariance structure of the experimental design. Environmental correlations estimated using both methods are nearly identical (S3 Fig). Genetic correlations estimated by correlation partitioning are sometimes slightly smaller in magnitude than those obtained by linear modeling (S3 Fig). This bias is conservative; it may prevent us from identifying cases in which the environmental and genetic correlations significantly differ but will not tend to create such cases. Despite this reduced power, we rely on the correlation-partitioning approach, which is substantially faster, because our goal is to estimate environmental and genetic correlations for thousands of trait pairs.

Unlike the mapping analysis, which considered phenotypes across all three classes of cell type (unbudded, small-budded, and large-budded), this correlation-partitioning analysis can only be applied to pairs of phenotypes that can be measured in the same cell. Three of the 36 pleiotropic QTL exclusively affect traits from different cell types. For example, a QTL on chromosome 4 affects the shape of the nucleus in unbudded cells as well as in large-budded cells. The correlation between these traits cannot be partitioned into a within-strain component because these traits are never measured in the same single cell. Excluding these three QTL leaves 33 pleiotropic QTL.

The 167 single-cell morphological features we measured represent 5,645 pairs of traits (378, 1,081, and 4,186 pairs of morphological features pertaining to unbudded, small-budded, and large-budded cells, respectively). Because some of these traits are related, these thousands of trait pairs are not independent. This dependence prevents us from reliably counting the absolute number of traits that are influenced by vertical versus horizontal pleiotropy. Still, partitioning correlations into a nongenetic ($r_W$) and a genetic ($r_B$) component for thousands of trait pairs enables us to (1) analyze a network describing the degree to which morphological traits are interconnected or modular and (2) detect examples of horizontal pleiotropy if they exist. This approach differs from that of previous studies of pleiotropy that used principal component analysis (PCA) to understand which traits are correlated [17]. Performing PCA on the individual-cell data is not the same as controlling for $r_W$ because PCA would ignore the strain groupings, which can then dominate the analysis (the classic "heterogeneous subgroup" problem in correlation analysis [72]). As a consequence, PCA can miss cases of horizontal pleiotropy and obscure, rather than reveal, inherent trait relationships.

## Inherent relationships between traits contribute to pleiotropy

We focus first on vertical pleiotropy by analyzing correlations that exist in the absence of any genetic differences ($r_W$). In the analyses that follow, when we refer to $r_W$ (or $r_B$), we mean the

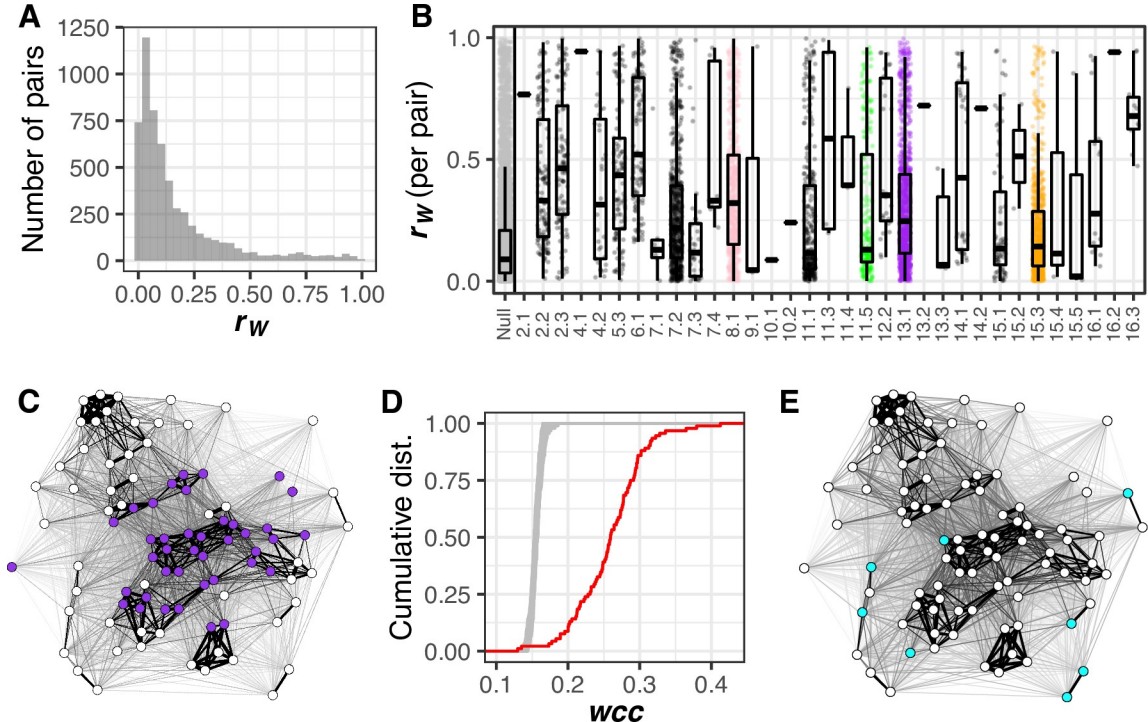

**Fig 2. Pairs of traits with high correlation across clones are overrepresented among those influenced by pleiotropic QTL.** Within-genotype correlations ($r_W$) are calculated for 5,645 pairs of morphological traits. (A) Histogram showing distribution of $r_W$. (B) Each point represents $r_W$ for a pair of traits. The null distribution displays $r_W$ for all 5,645 pairs of traits. Every other distribution displays $r_W$ for pairs of traits influenced by a single QTL. Horizontal axis labels represent the chromosome on which a QTL resides, followed by the order in which that QTL appears on the chromosome. Colored points correspond to those QTL highlighted in the same color in Fig 1A. Each box plot shows the median (center line), IQR (upper and lower hinges), and highest value within 1.5 × IQR (whiskers). (C) Force-directed network visualizing how pairs of morphological features correlate across clones. Each node represents a single-cell morphological trait measured in large-budded cells. For networks representing traits from unbudded and small-budded cells, see S4 Fig. The thickness of the line connecting each pair of nodes is proportional to $r_W$. Node position in the network is determined using the Fruchterman-Reingold algorithm. Purple nodes correspond to traits influenced by a QTL on chromosome 13 containing the *HOF1* gene. (D) Cumulative distributions ("dist.") of weighted clustering coefficients (*wcc*) in a network created using measured values of $r_W$ (red line) or in 100 permuted networks (gray lines) for traits corresponding to large-budded cells. Permutations were performed by sampling $r_W$, without replacement, and reassigning each value to a random pair of traits. For distributions summarizing *wcc* in networks representing traits from unbudded and small-budded cells, see S4 Fig. (E) The same network as in (C), with colored nodes corresponding to traits influenced by the leftmost QTL on chromosome 15. Data underlying this figure can be found at https://osf.io/b7ny5/. IQR, interquartile range; QTL, quantitative trait loci.

magnitude of the correlation, as the sign has no relevance for arbitrary pairs of traits. The distribution of $r_W$ across traits that are influenced by the same QTL reflects the degree to which that QTL acts via vertical pleiotropy. The overall pattern of $r_W$ values (i.e., whether there are isolated clusters of highly correlated traits versus a densely interconnected network of traits) reflects the modularity of the underlying biological system. These within-strain correlations are estimated with extremely high precision because of our large sample size of hundreds of thousands of clonal cells (800 per each of 374 strains).

Most pairs of single-cell morphological traits are not strongly correlated across clonal cells (Fig 2A). Median $r_W$ is $< 0.1$, and 74% of pairs have $r_W < 0.2$. Even if we allow for nonlinear correlations by transforming data using a nonparametric model that finds the fixed point of maximal correlation [73], $r_W$ is less than 0.2 for roughly 65% of pairs. These observations suggest that most of the morphological traits we surveyed are not inherently related; i.e., for any individual cell, the value of one trait does not predict well the values of most other traits.

Nonetheless, the distribution of $r_W$ has a prominent right tail (Fig 2A), indicating that some morphological features are strongly correlated across clonal cells. These correlated features are more likely to be influenced by pleiotropic QTL. Among pairs represented by this right tail (specifically, those with $r_W > 0.2$), 75% consist of traits that share at least one QTL influence; the same is true for only 36% of pairs with $r_W < 0.2$. These percentages are similar after changing our QTL-detection threshold to correct for having tested multiple phenotypes (66% and 21%, respectively). Further, the number of pleiotropic QTL influencing both traits in a pair correlates with that pair's $r_W$ (Pearson's $r$ is 0.52 before correction and 0.54 after). These results suggest that inherent correlations among morphological features often cause genetic perturbations to one feature to have consequences on another. In other words, we observe evidence of vertical pleiotropy.

Next, we studied each of the 33 pleiotropic QTL one at a time, asking whether they influence pairs of traits with higher $r_W$ than expected by chance. Most QTL have a higher median $r_W$ for the pairs of traits they influence than the median $r_W$ given by all possible pairs of traits (Fig 2B). This difference suggests that vertical pleiotropy drives a large portion of the pleiotropy we detect.

We also used network analysis to move beyond the pairwise comparisons in Fig 2A and ask if morphological traits tend to be clustered into modules. Traits with higher $r_W$ do indeed tend to group into clusters in networks in which the single-cell morphological traits are nodes and the $r_W$ magnitudes are edge weights (Fig 2C). This need not have been the case; pairs of traits with high $r_W$ could have been distributed throughout the network without necessarily being clustered near other high-$r_W$ pairs. Instead, networks representing single-cell morphological features demonstrate more clustering than do random networks drawn from the same values of $r_W$ (Fig 2D; for corresponding figures from unbudded and small-budded trait networks, see S4 Fig). This observation might indicate that morphological phenotypes have a modular organization, whereby phenotypes within a module exert influence on one another but exert less influence on phenotypes from other modules. However, this observation could also result from human bias when enumerating phenotypes that can be measured, in the sense that phenotypes that bridge modules might somehow be absent from the dataset. The comprehensive nature of CalMorph diminishes this concern. A related concern is that apparent modules are formed by trivially related phenotypes, such as the radius and diameter of a circular object, but we do not find such trivial relationships among the CalMorph phenotypes. Even a high correlation between the length and area of the nucleus implies a constraint on nuclear aspect ratio.

Our analysis of within-strain trait correlations so far suggests that natural variation contributing to variation in multiple single-cell morphological features often acts via vertical pleiotropy. Still, there are hints of another mechanism at play. Some QTL tend to influence traits that are not clustered in the correlation network (e.g., Fig 2E). And many pleiotropic QTL influence some pairs of traits with negligible $r_W$ (Fig 2B). To investigate how often pleiotropy is not predicted by the degree to which morphological features correlate in the absence of genetic variation, in the next section we compare trait correlations present across clones ($r_W$) to those present between genetically diverse strains ($r_B$).

## Many traits are more strongly correlated across strains than they are across clones

When genetic changes that perturb one trait have collateral effects on another, we expect the way traits correlate across genetically diverse strains to reflect trait correlations across clones (i.e., $r_B = r_W$). When this condition is met, pleiotropy can be viewed as an expected consequence of inherent relationships between traits, i.e., vertical pleiotropy. On the other hand, if a

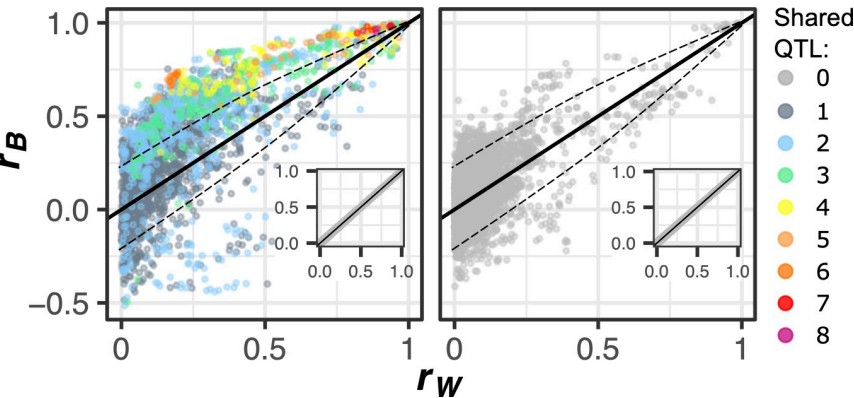

**Fig 3. Natural genetic variation affects the correlation between morphological features.** The absolute value of the between-strain correlation ($r_B$), made negative when $r_B$ and $r_W$ have opposite signs, is plotted against the absolute value of the within-strain correlation ($r_W$), for each pair of traits. The plot at left shows pairs of traits that share at least one QTL influence. The color of each point represents the number of pleiotropic QTL that influence both traits in that pair. The plot at right shows pairs of traits that share no QTL influence. The dashed line represents a Bonferroni-corrected significance threshold of $p < 0.01$. Insets represent the results of correlation partitioning performed after randomly assigning individual cells to groups (pseudo-strains) having the same numbers of cells as the actual strains. Data underlying this figure can be found at https://osf.io/b7ny5/. QTL, quantitative trait loci.

QTL influences two traits that do not correlate across clones, it may cause these traits to correlate across strains in which this QTL is segregating. In this case, we expect $r_B$ will be greater than $r_W$, suggesting horizontal pleiotropy.

After correcting for testing thousands of trait pairs, $r_B$ significantly exceeds $r_W$ in 25% of all trait pairs and 43% of pairs in which at least one pleiotropic QTL influences both traits (Fig 3; left panel; 43% of points are above the envelope, which represents a Bonferroni-corrected significance threshold of $p < 0.01$). This percentage grows to 53% in the smaller set of QTL that are detected after correcting for testing many traits. In the majority of cases in which $r_B$ significantly differs from $r_W$, $r_B$ is greater than $r_W$ (Fig 3; left panel; 78% of points outside the envelope are above it). The magnitude of the increase in $r_B$ versus $r_W$ tends to scale with the number of pleiotropic QTL that jointly influence both traits in a pair (Fig 3; left panel; colors get warmer farther above the envelope). These observations are consistent with the hypothesis that QTL acting via horizontal pleiotropy increase $r_B$ relative to $r_W$.

However, horizontal pleiotropy is not the only reason traits may correlate differently across strains versus across clones. We find significant deviations in $r_B$ relative to $r_W$ in 14% of pairs for which no pleiotropic QTL influence both traits (Fig 3; right panel) or 16% of such pairs when using the smaller set of QTL that are detected after correcting for testing many traits. This observation may suggest the presence of pleiotropic genetic variants that we did not have statistical power to detect in our QTL screen. But an alternate explanation for the observed increases in $r_B$ over $r_W$ is that perhaps we sometimes underestimate $r_W$.

One reason $r_W$ could be underestimated is that single-cell measurements are noisier than group-level averages. To test this possibility, we randomly assigned individual cells to groups (pseudo-strains) having the same number of cells as the actual strains and found that in these permuted data, $r_B$ and $r_W$ never significantly differ (Fig 3; insets). Because detection of $r_W$ was not underpowered relative to $r_B$, we conclude that measurement noise does not meaningfully obscure $r_W$. Another reason $r_W$ could be underestimated is if trait correlations across strains are more linear than those across clones. To test this possibility, for every pair of traits we transformed the single-cell trait measurements using a nonparametric model that finds their maximal correlation [73]. This transformation abrogated significant differences in $r_B$ relative

to $r_W$ for fewer than 5% of affected trait pairs. Another reason $r_W$ might be less than $r_B$ is if there tends to be less phenotypic variation within strains than between strains. Contrary to this prediction, every morphological trait we surveyed varies more within strains than between strains. Indeed, broad-sense heritability of the traits did not exceed 15% (S1 Fig), reflecting that within-strain phenotypic variation (e.g., variation in a cell's progress through the division cycle) accounted for at least 85% of the total variation. A final reason $r_W$ could be poorly estimated is if nongenetic heterogeneity across different subpopulations within clonal populations causes variation in $r_W$. Therefore, next we investigated whether the relationship between single-cell features varies for clonal cells in different stages of the cell-division cycle.

## Inferring a cell's progress through division from fixed-cell images

Pairs of traits for which $r_B$ is strong whereas $r_W$ is not should reflect horizontal pleiotropy, but a closer examination of some of these pairs revealed traits that should correlate because of simple geometric constraints. For example, cell size and the width of the bud neck should correlate because of the constraint that, even at its maximum, bud neck width cannot be larger than the diameter of the cell. When measured in small-budded cells, these two traits are correlated across yeast strains ($r_B = 0.40$) but are significantly less correlated across clones ($r_W = 0.15$). Given the simple geometric constraint coupling the width of the bud neck to the cell's size, why is there a discrepancy between $r_B$ and $r_W$? We reasoned that this discrepancy exists because the correlation between cell size and neck width is disrupted during particular moments of cell division; e.g., the width of the bud neck starts small even for large cells (Fig 4A; cell micrographs outlined in blue show two cells in the progress of budding). If the relationship between morphological features varies during cell division, $r_W$ may represent a poor summary statistic.

How often does the relationship between morphological traits change during cell division? Our single-cell measurements are primed to address this question: we fixed cells during exponential growth and imaged hundreds of thousands of single cells, thereby capturing the full spectrum of morphologies as cells divide. A remaining challenge is sorting these images according to progress through cell division and then remeasuring the correlation between morphological features within narrow windows along that progression.

We performed this sorting using the Wishbone algorithm [74]. This algorithm extracts developmental trajectories from high-dimensional phenotype data (typically single-cell transcriptome data). We applied Wishbone separately to cells belonging to each of the three cell types defined by morphometric analysis (unbudded, small-budded, and large-budded cells). The trends describing how morphological features vary across Wishbone-defined cell-division trajectories are consistent with previous observations of how morphology changes as yeast cells divide [75,76] (Fig 4A; line plots). For example, Wishbone sorts fixed-cell images in such a way that cell area increases throughout the course of cell division (Fig 4A; upper left panel), and nuclear elongation occurs just before nuclear division (Fig 4A; lower left panel). These trajectories also match our own observations of how morphological features change as live cells divide, which we tracked by imaging at 1-minute intervals one of the 374 progeny strains that we had engineered to express a fluorescently tagged nuclear protein (HTB2–green fluorescent protein [GFP]) (Fig 4A; micrographs). We chose this particular strain because it does not deviate from the average morphology of all 374 recombinants by more than 1 standard deviation for any of the phenotypes we measure.

To further validate Wishbone's performance, we asked whether it could reconstruct the time series of live-cell images from the HTB2-GFP strain. We obtained time series for 78 single dividing cells, each imaged over at least 20 time points. Quantifying morphological phenotypes

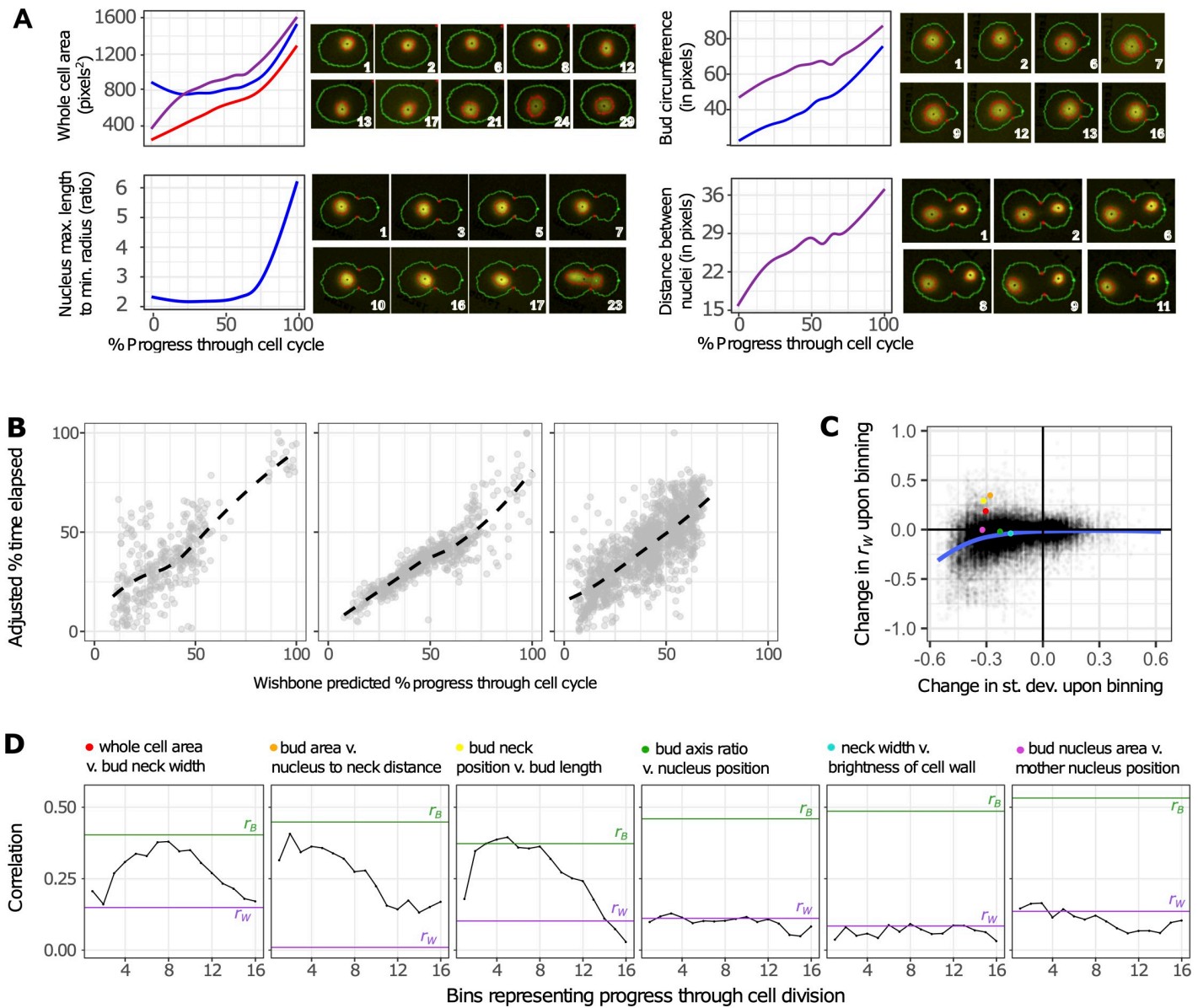

**Fig 4. Morphological features vary as cells divide.** The morphological features of unbudded (red), small-budded (blue), and large-budded (purple) cells change as these cells progress through the cell cycle. (A) Variation of four traits through the cell cycle. Line plots represent fixed-cell images from all 374 mapping-family strains, positioned on the horizontal axis based on progression through the cell cycle as calculated by Wishbone [74]. Regression lines are smoothed with cubic splines, calculated with the "gam" method in the R package ggplot2 [99], to depict trends describing how each displayed trait varies across the estimated growth trajectory. The displayed trends match those observed in micrographs of live cells progressing through division. Each series of micrographs displays a different live cell imaged over several minutes, which are displayed in the lower right corner of each micrograph. (B) Centered data for 11, 23, and 44 unbudded, small-budded, and large-budded cells, respectively, show how Wishbone sorts live cells in a way that recapitulates the actual time series. Each point in these plots represents a cell image from a single time point. The horizontal axis represents Wishbone's estimation of how far that cell has progressed through division. The vertical axis displays time as a percentage of the total time elapsed and adjusted in a way that controls for every cell having started at a different place in the cell division cycle at time zero. Trend lines are smooth fits using the "loess" method in the R package ggplot [99]. (C) The correlation between some morphological features changes throughout the course of cell division. The scatterplot shows how binning influences both the phenotypic correlation (vertical axis) and phenotypic variation (horizontal axis) across clones. Each point represents these values for a pair of traits as measured in 1 of 16 bins. The value on the horizontal axis represents whichever trait in each pair had the larger decrease in standard deviation ("st. dev."), as such decreases are likely to reduce the correlation on the vertical axis. The blue line shows a smooth fit by loess regression. Colored points on the scatterplot correspond to bin 5 for each pair of traits represented by the line plots in (D). (D) These line plots show three pairs of traits for which binning increases $r_W$ such that it approaches $r_B$ (leftmost three plots), and three pairs of traits for which $r_W$ does not approach $r_B$ even after binning (rightmost three plots). In each plot, $r_B$ is shown as the horizontal green line, $r_W$ (without binning) is shown as the horizontal purple line, and $r_W$ for each bin is shown in black. Data underlying this figure can be found at https://osf.io/b7ny5/. max., maximum; min., minimum.

from live-cell images in a high-throughput fashion proved difficult because the morphometric software was optimized for fixed-cell images, and as cells grow and bud, the cells and their nuclei can move out of the focal plane. Also, although we used short exposure times when imaging GFP fluorescence, there are concerns about phototoxicity and associated growth and morphology defects [77]. For these reasons, we expect Wishbone to perform better on fixed-cell images than on time series of live-cell images. Still, Wishbone's cell-division trajectories recapitulate the time course. When we align time-series data across live cells by centering on each cell's average predicted progress through division, Spearman's $r$ is 0.65, 0.91, and 0.77 for time series corresponding to each of the three cell types (Fig 4B; see S5 Fig for recapitulation of 78 individual time series). These correlations are substantially higher than those obtained by repeating the merging procedure after randomly permuting each time series (corresponding Spearman's $r$ of 0.42, 0.43, and 0.56). These observations suggest that Wishbone is effective at properly assigning single-cell images to their position in the cell cycle.

## Cell-cycle state can influence the relationship between morphological features

To identify cases in which significant differences in $r_B$ versus $r_W$ might result because $r_W$ is sensitive to cell-cycle state, we first assigned each imaged yeast cell from the QTL-mapping population to one of 16 equal-sized bins based on Wishbone's estimation of how far that cell had progressed through division. Because we did this separately for each of the previously defined cell stages (unbudded, small-budded, and large-budded), this additional binning finely partitions cell division into 48 ($16 \times 3$) stages. To hold genotype representation constant across each of the 48 bins, we performed binning separately for each of the 374 mapping-family strains, then merged like bins across strains. We then performed correlation partitioning on each bin separately.

Binning cells by cell-cycle state typically decreased the amount of phenotypic variation per bin, which we expect in turn to obscure the correlation between traits. Consider an extreme example: if there is no phenotypic variation remaining for a given trait, it cannot covary with any other traits. Indeed, for most pairs of traits, the binning procedure either decreases $r_W$ or does not have a dramatic effect on it; decreases in $r_W$ are especially evident for trait pairs in which variation of at least one of the traits shows a relatively large decrease upon binning (Fig 4C). However, for some pairs of traits, despite the decrease in phenotypic variation for at least one trait, the correlation between traits improves upon binning. For example, binning by cell division increases the correlation between cell size and the width of the bud neck (Fig 4D; left-most plot) such that it approaches $r_B$. This increased correlation is consistent with our hypothesis that the process of cell division was obscuring the dependency of bud neck width on cell size. Examining more pairs of traits for which binning tends to increase $r_W$ (Fig 4C; red, orange, and yellow points) reveals additional cases in which the process of cell division decouples traits that are otherwise correlated and in which binning reveals the underlying correlation (Fig 4D; leftmost three plots).

Despite the evidence that cell asynchrony alters some trait correlations, many cases remain in which heterogeneity in cell-cycle state does not explain the observed discrepancy between $r_W$ and $r_B$ (Fig 4D; rightmost three plots). We previously demonstrated that $r_B$ significantly exceeds $r_W$ in 24% of all trait pairs (1,389/5,645) (Fig 3). For almost half of these pairs (689 pairs), binning by cell division does not resolve the discrepancy between $r_B$ and $r_W$ to any extent; in other words, $r_W$ does not increase in any of the 16 bins. For an additional 193 pairs, binning by cell division resolves the discrepancy by at most 5% in any bin. These results imply that cell-cycle heterogeneity does not cause the discrepancy between $r_W$ and $r_B$ in the majority

of cases, suggesting that the elevation of $r_B$ over $r_W$ could be explained by QTL demonstrating horizontal pleiotropy.

## Many QTL demonstrate horizontal pleiotropy

To test horizontal pleiotropy further, we asked whether pleiotropic QTL cause increases in $r_B$ relative to $r_W$. Not all pleiotropic QTL affect pairs of traits for which $r_B$ is significantly greater than $r_W$, so we focus this analysis on the 27 out of 33 QTL that do. We divided our yeast strains into sets in which a given QTL is not segregating, then remeasured the difference between $r_B$ and $r_W$. More specifically, for each QTL, we split the 374 phenotyped yeast strains into two groups based on whether they inherited the wine or the oak parent's allele at the genotyped marker closest to the estimated QTL location. Then we repeated correlation partitioning on each subset of strains and compared the results to those obtained from the complete set. For each QTL, we focused on trait pairs in which (1) both traits are affected by this QTL and (2) $r_B$ is significantly greater than $r_W$. Across all such pairs, median $r_B$ tends to decrease upon eliminating allelic variation at the marker nearest the QTL (Fig 5A). No similar reduction in $r_B$ is observed when we focus on pairs of traits that are not affected by each QTL (Fig 5A) and no similar reduction is observed in $r_W$ (median reduction in $r_W$ is 0.0001).

There appear to be two ways in which a QTL may affect $r_B$. In some cases, eliminating genetic variation at the marker nearest a QTL decreases $r_B$ in both resulting subpopulations. Such cases are consistent with a straightforward scenario in which horizontal pleiotropy results when a QTL that influences two or more traits (that are otherwise weakly correlated) is segregating in a population (Fig 5B; top row shows that the correlation is strongest in the mixed population where both oak and wine alleles are segregating). In other cases, eliminating allelic variation at a QTL site decreases $r_B$ in only one of the two resulting subpopulations (i.e., the subpopulation possessing either the oak or the wine allele). This observation demonstrates that horizontal pleiotropy can emerge by virtue of a QTL allele strengthening a correlation between two traits so that genetic variation affecting one trait is more likely to affect the other when that allele is present [78,79] (Fig 5B; bottom row).

How many cases in which $r_B$ significantly exceeds $r_W$ can be explained, to some extent, by horizontal pleiotropy (i.e., a QTL increasing the between-genotype correlation)? For every trait pair in which $r_B$ significantly exceeds $r_W$ and at least one QTL influences both traits in the pair (1,108 pairs total), eliminating allelic variation at the marker nearest at least one of the shared QTL causes $r_B$ to decrease in one or both of the resulting subpopulations (Fig 5C: solid black line in rightmost plot). About 60% of these decreases affect both subpopulations (e.g., Fig 5B; top row) and 40% affect only one subpopulation (e.g., Fig 5B; bottom row). These decreases in $r_B$ appear to resolve the discrepancies in $r_B$ versus $r_W$ more often and to a greater extent than does accounting for cell-cycle heterogeneity (Fig 5C; leftmost plot). Some QTL have larger impacts on $r_B$ than do others (Fig 5C). Eliminating allelic variation near a QTL on chromosome 13 decreases $r_B$ in the largest number of traits pairs (658). Subtracting the influence of a QTL on chromosome 15 decreases $r_B$ to the greatest extent; the average decrease across 342 affected trait pairs is 0.07.

One caveat that remains is whether these examples of pleiotropy represent single genetic changes that influence multiple traits or the presence of multiple nearby genetic variants within each QTL. Our finding that there are two types of horizontal pleiotropy (Fig 5B) provides some insight. The first type of horizontal pleiotropy (Fig 5B; upper row) may result from the presence of multiple genetic variants segregating together because recombination has not broken them apart. However, the second type of horizontal pleiotropy (Fig 5B; lower row) cannot be explained in the same way because a strong trait correlation exists in the absence of

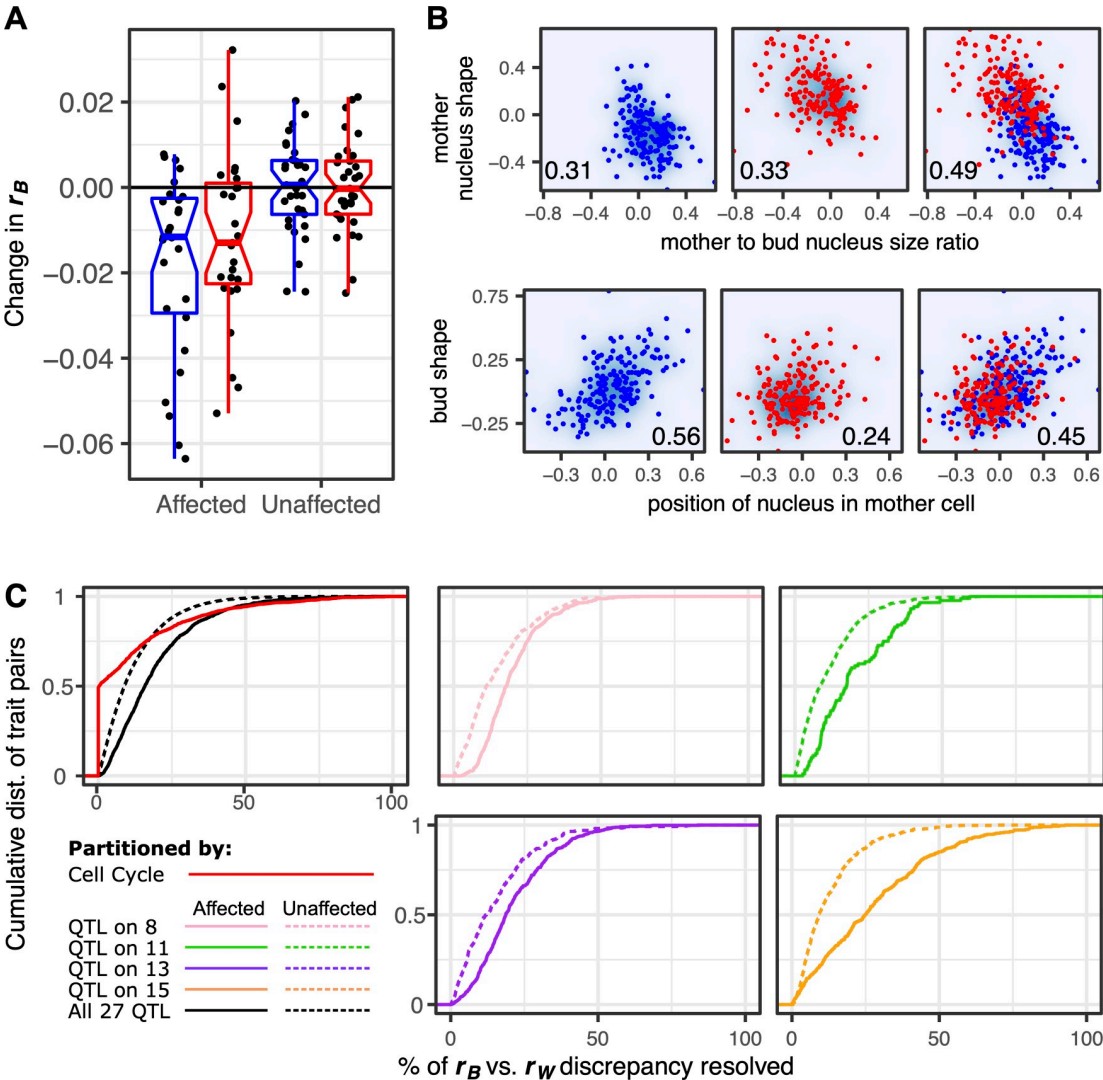

**Fig 5. Many QTL demonstrate horizontal pleiotropy.** (A) Eliminating allelic variation at the site of each QTL tends to reduce $r_B$. The vertical axis represents how $r_B$ changes upon eliminating allelic variation at each QTL site. Each point represents the median change in $r_B$ for all pairs of traits that are affected or unaffected by one of the 27 QTL suspected of horizontal pleiotropy. Box plots summarize these changes in $r_B$ when remeasured across strains possessing the wine (red) or the oak (blue) allele at the marker closest to the QTL. (B) The upper and lower series of three plots demonstrate two different ways that a QTL can increase the correlation between traits. Each point represents a yeast strain possessing either the wine (red) or the oak (blue) allele at a marker closest to a QTL on chromosome 15 (upper) or 8 (lower). In the upper plots, the QTL increases the correlation between nucleus shape and size ratio when it is segregating across strains. In the lower plots, the oak allele strengthens a correlation between bud shape and the position of the nucleus in the mother cell that is weak in the wine subpopulation. Numbers in the lower corner of each plot represent $r_B$ for the strains displayed. (C) Cumulative distributions ("dist.") display the extent to which binning cells or splitting strains resolves the difference between $r_B$ and $r_W$. When calculating percent resolved (horizontal axes), we always plot the value in whichever subset (e.g., wine or oak) this percent is greatest. If subsetting always worsens the discrepancy between $r_B$ versus $r_W$, we score this as 0% resolution. Only pairs of traits for which $r_B$ is significantly greater than $r_W$ are considered. The pink, green, purple, and orange lines show the effect of splitting strains by whether they inherited the wine or oak allele at the marker closest to each of four QTL (colors correspond to QTL in Fig 1A). In these plots, comparing the solid versus dotted lines shows that splitting strains resolves the discrepancy between $r_B$ and $r_W$ more often for pairs in which both traits are affected by the QTL than pairs in which both traits are unaffected. The black lines in the leftmost plot summarize these effects across 27 QTL, displaying for each trait pair the largest resolution in the $r_B$ versus $r_W$ discrepancy observed across all QTL that affect the pair of traits (solid line) or all QTL that do not (dotted line). The red line shows the effect of binning cells by their progress through division, displaying the largest resolution in the $r_B$ versus $r_W$ difference across all 16 bins. Data underlying this figure can be found at https://osf.io/b7ny5/. QTL, quantitative trait loci.

allelic variation at that QTL (blue points). Therefore, although multiple closely linked variants might underlie the difference in the trait correlation between the oak and wine alleles of the QTL, they would act at the level of the correlation rather than of individual traits.

## Spontaneous mutations alter the relationships between morphological features

Our finding that some QTL alleles appear to strengthen correlations between otherwise weakly correlated traits (Fig 5B; lower panel) lends credence to the idea that the relationships between phenotypes, and thus the extent of phenotypic modularity (or integration), are mutable traits [80]. This finding has implications for evolutionary medicine, in particular evolutionary traps, e.g., strategies to contain microbial populations by encouraging them to evolve resistance to one treatment so that they become susceptible to another [40–42]. These traps will fail if targeted correlations can be broken by mutations. To test whether spontaneous mutations can alter trait correlations, we analyzed the cell-morphology phenotypes of a collection of yeast MA lines [55]. These MA lines were derived from repeated passaging through bottlenecks, which dramatically reduced the efficiency of selection and thereby allowed retention of the natural spectrum of mutations irrespective of effect on fitness [56]. We previously imaged these lines in high throughput (>1,000 clonal cells imaged per each of 94 lines) [51].

Because MA lines contain private mutations unique to each strain, they are not amenable to QTL mapping and between-strain trait correlations have less meaning. Instead, we focused on within-strain correlations, which we expected to be consistent across strains because of the limited number of mutations distinguishing the strains (an average of 4 single-nucleotide mutations per line [56]), except if a rare mutation does indeed alter the correlation. To determine if such correlation-altering mutations exist, we calculated within-strain correlations for each strain separately and asked, for each trait pair, whether any strains had extreme correlations relative to the other strains. For most trait pairs, the MA lines' trait correlations did not vary much from each other or from that of the ancestor strain (Fig 6). However, in several instances, we observed a trait-pair correlation dramatically outside the range of the other trait pairs and more than 4 standard deviations from the mean (Fig 6A). Some mutations appear to influence many trait-trait relationships (mutations found in blue- and purple-colored strains in Fig 6B and 6C), whereas others influence fewer (mutations found in magenta-colored strain in Fig 6C). Mutations that alter trait correlations are not necessarily the result of rare events such as aneuploidies or copy number variants; all five strains highlighted in Fig 6 do not possess these types of mutations and instead possess at least one single-nucleotide mutation [56].

Given that in the small sampling of spontaneous mutations captured by the MA strain collection, we found several that appear to alter the relationship between morphological features, we think such mutations are common enough to merit further consideration in evolutionary models. The mutations in the outlier lines provide candidate correlation-altering mutations for future mechanistic studies as well.

## Different environments alter the relationships between morphological features

We have used nongenetic heterogeneity within clonal populations to uncover inherent trait correlations. One might consider achieving the same aim by using instead the nongenetic perturbations represented by different environmental treatments. However, our results suggest that trait correlations can be highly context dependent, changing across cell-cycle state (Fig 4) and genetic background (Fig 5B lower panel and Fig 6). If trait correlations change across environments, then the intricacies of the environment-specific effects would need to be

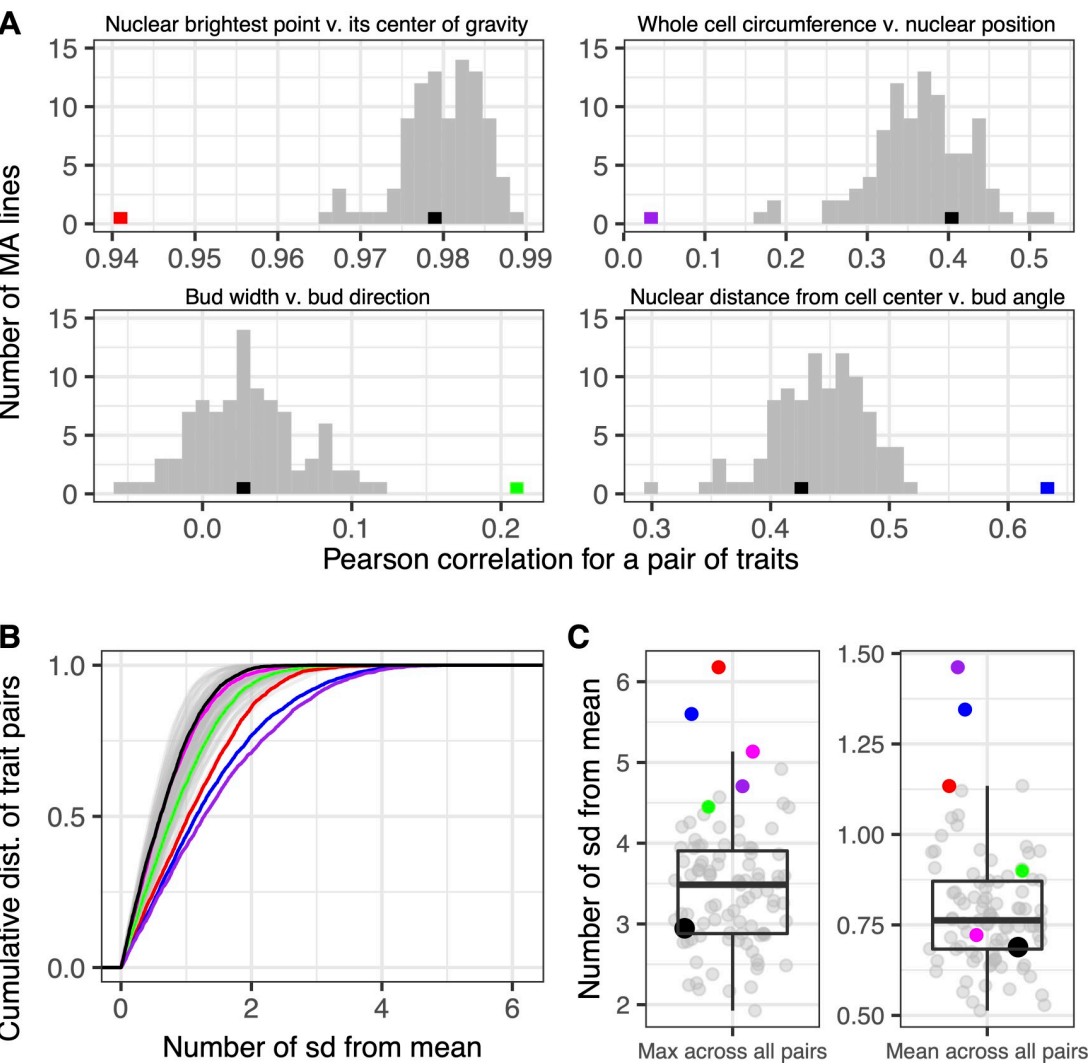

**Fig 6. Some MA lines display unique relationships between certain pairs of traits.** In all plots, black represents the ancestor of the MA lines and colors represent MA lines with trait correlations that differ from other lines (strains: black = HAncestor, green = DHC81H1, red = DHC41H1, magenta = DHC40H1, blue = DHC66H1, purple = DHC84H1; see S2 Table in [51]). (A) Histograms display the number of MA lines with Pearson correlations corresponding to the values on the horizontal axis for four example pairs of traits; the number of bins is set to 30. (B) This plot displays, for each of the 94 MA lines, the cumulative distribution ("dist.") of the number of sd away from the mean correlation across all trait pairs. (C) Plots display, for each MA line, the maximum deviation from the mean observed for any pair of traits (left) and the average sd observed across all pairs of traits (right). Data underlying this figure can be found at https://osf.io/b7ny5/. MA, mutation-accumulation; sd, standard deviation.

incorporated into any inferences about vertical and horizontal pleiotropy, adding a complicating dimension to the analysis.

To investigate the potential utility of across-environment trait correlations for distinguishing horizontal from vertical pleiotropy, and to further explore the context dependence of trait correlations, we analyzed trait correlations across a range of concentrations of the Hsp90-inhibiting drug geldanamycin (GdA). We showed previously that GdA affects cell morphology [51], so it presents an opportunity to analyze how correlations among these traits vary across environments. We performed this analysis using a subset of the yeast strains from our QTL-mapping family (S6 Fig), partitioning trait correlations into a pooled within-strain component ($r_W$) and a between-strain component ($r_B$).

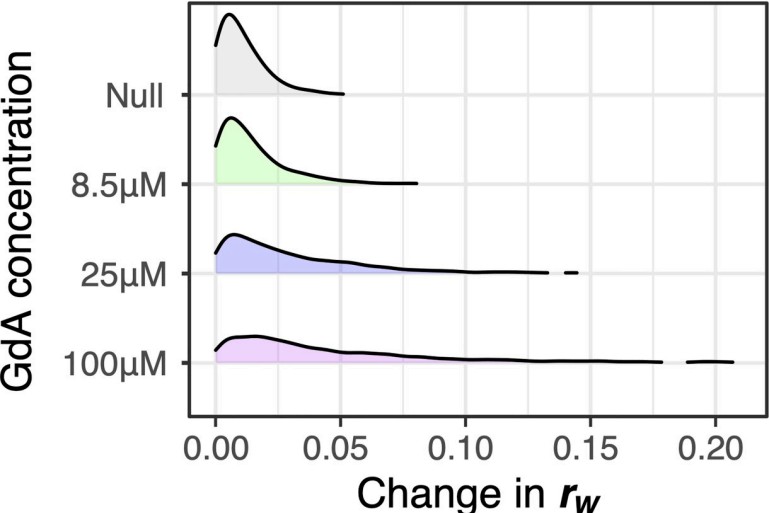

**Fig 7. The correlations between traits changes depending on drug concentration.** Plots display the density of trait pairs for which the within-strain correlation ($r_W$) changes by the amount shown on the horizonal axis. To calculate how each drug treatment changes $r_W$, we subtracted $r_W$ observed for a pair of traits in the drug condition from that observed in a paired experiment that lacked the drug. The absolute value of this change is displayed. These changes are smallest in the null condition, which represents the change in $r_W$ observed across replicate experiments lacking the drug. For clarity, we exclude regions of the plot for which density is less than 0.01. Data underlying this figure can be found at https://osf.io/b7ny5/. GdA, geldanamycin.

GdA alters the correlations between morphological traits. The impact of GdA on $r_W$ increases with the concentration of GdA (Fig 7), suggesting that more extreme environmental differences are more likely to result in changes in $r_W$. We conclude that looking across diverse environments is not a good way to understand the inherent relationships between traits that exist in a single environment. Indeed, previous studies of pleiotropy have treated growth parameters in different environments as different traits [81] rather than as a way to estimate inherent trait correlations.

## Discussion

Although evolutionary biologists and medical geneticists alike appreciate that organismal traits can rarely be understood in isolation, the extent and implications of pleiotropy have remained difficult to assess. A common approach to measuring pleiotropy has been to count phenotypes influenced by individual genetic loci [18,34,35]. For example, the median number of skeletal traits affected per QTL in a mouse cross was six (out of 70 traits measured); this small median fraction of traits suggests that variation in skeletal morphology is modular [17,31]. Of course, for a count of traits to be meaningful, the full trait list must be comprehensive, and correlations between traits must be properly accounted for [18,34,35]. We aimed for comprehensiveness in a very similar way to the studies of mouse skeletal traits, by systematic phenotyping of a large number of morphological traits. However, we addressed the need for a principled approach to separating inherent trait correlations from those induced by genetic differences in a new way: by extending the analysis to include within-genotype correlations and thereby enabling an operational definition of the distinction between vertical and horizontal pleiotropy.

Our comprehensive analysis of how thousands of trait pairs covary within and between mapping strains yields an unprecedently quantitative and nuanced view of pleiotropy. We found support for modularity not only in the low median number of traits affected per QTL (five out of 167) but also in the way that within-genotype correlations grouped traits into

relatively isolated clusters (Fig 2). We also found ample evidence of horizontal pleiotropy layered on top of that modularity, with many cases of between-genotype trait correlations that exceeded within-genotype correlations (Fig 3).

Our results do not speak directly to whether modularity results from selection against pleiotropy in nature because we survey only two natural genetic backgrounds (wine and oak). In other words, the presence of modularity is not necessarily evidence that it is adaptive or that it is maintained by natural selection. Future work comparing MA lines to a larger collection of natural isolates might help answer questions about the extent to which selection purges pleiotropic mutations.

Our partitioning of between-strain (genetic) and within-strain (environmental) correlations relates to another approach to understanding trait interdependencies, the estimation of the so-called G matrix. This genetic variance-covariance matrix summarizes the joint pattern of heritable variation in a population of the traits that compose its rows and columns and is central to understanding how trait correlations constrain evolution. The G matrix arises in the multivariate breeder's equation, which describes the responses to selection of correlated traits [82]. If breeding is the goal, the distinction between vertical and horizontal pleiotropy is not so important, because both can impede selection. Indeed, any philosophical concern about what constitutes a biologically meaningful trait is irrelevant to the breeder, who actually cares about particular traits (e.g., milk yield and fat content).

G matrices are relevant not only to breeders but to evolutionary biologists as well, and it is worthwhile to place our results into this context. A major evolutionary question in the G-matrix literature is whether the G matrix itself can evolve. In other words, do short-term responses to selection (as captured in the breeder's equation) predict long-term responses or do constraints shift through time, perhaps in a way that facilitates (or is part of) adaptation [83]? Our results with MA lines add to evidence that the G matrix readily changes [84], in that individual mutations have major effects on particular trait correlations (e.g., Fig 6A). Our QTL-mapping results also support this view, in that some cases of horizontal pleiotropy appear to be caused by alleles that alter trait correlations (e.g., Fig 5B; bottom panel).

Another prominent question in the G-matrix literature is the extent to which the P matrix, which includes all sources of phenotypic variation and covariation, predicts the G matrix, which only includes additive genetic effects (i.e., those that respond to selection). If P predicts G well, as proposed by Cheverud [85], then inference of selection responses from patterns of trait covariation in a population would suffice when genetic analysis would be difficult or costly. Our results do not speak directly to this question, because we did not estimate G itself and instead estimated genetic correlations that include nonadditive effects. However, our results are informative from another angle, which is the comparison of genetic and environmental correlations. As we showed (Fig 3), although there are cases in which the environmental and genetic correlations have different signs, the environmental correlations do tend to match the signs of the genetic correlations and predict their magnitudes to some extent as well, consistent with similarity between P and G. Future experiments using clones embedded in a more complicated crossing scheme could properly partition P into G, E, and the nonadditive genetic components, to address Cheverud's conjecture [85] more directly. There are only a few reports of comparisons of E matrices [86], but we encourage increased attention to the E matrix to understand inherent trait correlations and to contextualize G in a way that diminishes concerns about which traits are biologically meaningful and therefore merit status as the matrix's rows and columns.

A major and unforeseen conclusion of our work is the extent to which context is crucial. We have shown that trait correlations change through the cell-division cycle, in different genetic backgrounds, and across a drug gradient. It is likely that macroenvironmental

differences alter trait correlations as well [87]. These findings provide insight as to how biological systems appear to be modular, as evolutionary theory predicts [24–29], yet in other studies appear to be highly interconnected [13,14]. Our results suggest that biological systems are modular but that these modules change across contexts such that the potential phenotypic impacts of a genetic change can be extensive.

These results support the idea that predicting the phenotypic impact of a genetic change requires a paradigm shift [88,89], away from merely mapping the relationships between traits and toward unfurling the range of contexts across which those relationships persist. Future work in this direction will not only advance understanding of the evolution of complex traits but will have practical benefits. For example, our approach demonstrates a potentially fruitful way to design evolutionary traps: studying within-genotype correlations across contexts to identify particularly immutable correlations between traits.

## Materials and methods

### Measuring the morphology of single yeast cells

Recombinant yeast strains were generated from a cross between the oak parent (BC233: *SPS2*:*EGFP*:*kanMX4/SPS2*:*EGFP*:*kanMX4*) and the wine parent (BC240: *SPS2*:*EGFP*:*natMX4/SPS2*:*EGFP*:*natMX4*), then genotyped at 225 markers in a previous study [54,60]; each resulting recombinant strain is a homozygous diploid. We prepared yeast cells from these strains for microscopy using published methods [50–52,90]. Briefly, yeast strains were grown in minimal media with 0.08% glucose in 96-well plates [91], harvested during exponential phase, fixed in 4% paraformaldehyde, stained for cell-surface manno-protein (with FITC–concanavalin A) and nuclear DNA (with DAPI), sonicated, mounted on 96-well glass-bottom microscopy plates, and imaged with a Nikon Eclipse TE-2000E epifluorescence automated microscope using a 40× objective and appropriate fluorescence filters. Three biological replicate experiments were performed, typically yielding a total of between 500 and 1,000 imaged cells per strain (S2 Fig).

### Statistical analysis and processing of cell image data

Cell image processing was performed similarly to previous studies [50–52,90]. Imaged cells were analyzed for quantitative morphological traits using the CalMorph software package [53], which reports on hundreds of morphological features that are each specific to one of three cell types: unbudded, small-budded, and large-budded cells. We excluded phenotypes for which >10% of cells had missing values, leaving 167 morphological features. Any cell that was not scored for all features pertaining to its type was eliminated. Each morphological trait was transformed via a Box-Cox transformation of the raw data with the value of lambda that makes the residuals of a linear regression of phenotype on strain most normal using the EnvStats package in R [92]. Internal controls (several wells representing the wine and oak parents) were present on every 96-well plate and were used to correct for effects on phenotypic variation that resulted from differences among replicate experiments, such as differences in the brightness of the cell stain. We calculated the mid-parent value for each phenotype on every plate, then calculated the average mid-parent value across all plates. For each phenotype, we found the difference between the plate-specific mid-parent value and the average mid-parent value across all plates. Then we subtracted this difference from each plate for the corresponding phenotype. After correction, any cell with a morphological feature that deviated from the average by more than 5 standard deviations was then eliminated, as investigation of such cells typically revealed these were CalMorph miscalls or cellular debris.

## QTL mapping

QTL interval mapping was performed similarly to previous studies [63] using the R/qtl package [64]. We performed a QTL scan using the function "scanone," which finds at most one QTL per chromosome, followed by the "scantwo" function, which allowed us to identify potential second additive QTL per chromosome. The yeast strains, which are homozygous diploids, were modeled as doubled haploids, and QTL models were fit using Haley-Knott regression. When comparing QTL across traits, QTL more than 5 cM apart on the same chromosome were counted as separate QTL. We estimate that a region of 5 cM contains on average eight genes, since there are 6,746 genes in the yeast genome [67], and the map length we calculated using R/qtl is 4,076 cM. In some cases, we detected a QTL in between two others on the same chromosome and within 5 cM of both. In these cases, there were typically many QTL found within a narrow region without any gaps of greater than 2 cM. We counted these as single QTL that affect many traits. Using this method, the largest QTL we detect spans 17 cM. A summary of all significant QTL effects, including their chromosomal locations in cM and which QTL on the same chromosome we considered unique, is provided in S1 Table (also see Fig 1A).

To determine significance thresholds, we employed the method of [65] as implemented in R/qtl [64]. QTL were assigned a *p*-value based on a trait-specific empirical distribution of genome-wide LOD score maximums from 10,000 (1,000 for the two-dimensional scan) randomly permutated datasets. We used a *p*-value cutoff of 0.05 to determine significant QTL. We also obtained a more-stringent set of significant QTLs by correcting for the testing of multiple traits by controlling the false-discovery rate across phenotypes. For each trait, we took the position and *p*-value of the maximum LOD score on every chromosome. We calculated q-values for this set of loci using the R "qvalue" package [93] and used a 0.05 q-value threshold to call significant QTL.

## Candidate gene swaps

All yeast transformations were performed using the lithium acetate [94] and *delitto perfetto* [66] methods. For each candidate gene, the gene was first deleted from haploid variants of both the wine and oak parental strains and replaced with a selectable marker, the yeast gene encoding orotidine-5′-phosphate decarboxylase (*URA3*). Gene knockouts were confirmed by growth on plates lacking uracil and DNA sequencing of the affected region. Next, the *URA3* selectable marker was replaced with the other parent's version of the candidate gene. These candidate gene "swaps" were selected by growth on 5-fluoroorotic acid and confirmed by sequencing of the affected region. For each candidate gene, we swapped a region containing the coding sequence plus 5–750 bp up- and downstream. We used the following regions of homology to define the boundaries of each swapped segment:

- Approximately 300 bp upstream of *PXL1*: TTATAATTGTGGTTTAGCGTTTCATAGTCGC

- Approximately 300 bp downstream of *PXL1*: CCTTATTCTCTATTCTTAGGCTCCTGTTCC

- Approximately 5 bp upstream of *HOF1*: GAAAGAATGAGCTACAGTTATGAAGCTTG

- Approximately 300 bp downstream of *HOF1*: GTATTCGTAACAAGTGACTCTAATGATAT

- Approximately 750 bp upstream of *RAS1*: CGACTAAAGGAATTATACCATCATGCATC

• Approximately 300 bp downstream of *RAS1*: GCATTTCTAAAAACAGAGCTTTTGCCG

These regions of homology were chosen by searching for regions of higher GC content nearby the start and end of each gene's coding sequence. In addition, we attempted to swap the wine and oak parents' versions of the *GPA1* gene on chromosome 8. Despite trying various regions of homology, we could not successfully replace *GPA1* with the *URA3* selectable marker in the oak parent. *GPA1* is known to be essential in some genetic backgrounds [95].

Though the recombinant strains we studied are homothallic diploids, the strains in Fig 1B (both the parental strains and the strains possessing the gene swaps) are haploid. Because the analyses in Fig 1B compare pairs of strains (e.g., the oak haploid parent to the wine haploid parent, or the wine haploid parent to the wine haploid parent possessing the oak allele of *PXL1*), we only considered experiments in which both strains in the pair were imaged in the same replicate experiment. To account for differences among replicate experiments, for each phenotype, we subtracted the value in one strain from the value in the other to calculate the phenotypic difference between strains in that replicate experiment; the reported value is the average of these differences across replicate experiments (S2 Table, Fig 1B).

## Calculation of correlation coefficients

We used WABA II as implemented in the multilevel package in R [71] to calculate cell-level ($r_W$) and strain-level ($r_B$) Pearson correlation coefficients for each pair of traits. We used an r-to-z transformation to determine whether differences in $r_B$ versus $r_W$ are significant at a Bonferroni-corrected *p*-value of 0.01 (this is a z-score cutoff of 4.63, given 5,645 pairs of traits were tested). To assess whether correlations across single cells generally result in different values than correlations across group-level averages, we assigned yeast cells to groups (pseudo-strains) randomly, maintaining the same number of cells per strain as in the actual data. To assess whether results would differ if we allowed for nonlinear correlations, we transformed the single-cell data using a nonparametric model that finds the fixed point of maximal correlation, implemented in the R package acepack [73]. To assess whether results from WABA differed from those obtained using a standard quantitative genetics model (S3 Fig), we implemented the latter using the nlme package in R [96] to specify a mixed-effects model with cells nested within strains. We specified a covariance structure that allows covariance between two traits but no covariance between cells or between strains. We used this model to calculate the environmental and genetic correlations for 350 pairs of randomly chosen traits.

## Live imaging single cells as they divide

For live imaging the morphology of dividing yeast cells, we chose one of the recombinant yeast strains, F2_292. This strain was chosen because it does not deviate from the average morphology of all 374 recombinants by more than 1 standard deviation for any of the phenotypes we measured. F2_292 was transformed to express a fusion protein of GFP and a nuclear protein (histone H2B encoded by *HTB2*). Two independent transformants were imaged in the GFP channel (for nuclei) and in brightfield (for cell outlines). We prepared live cells for imaging following published methods [91,97,98], in a similar way to that described above, except cells were neither fixed nor stained. Cells were taken during mid-log phase growth, seeded in 96-well glass-bottom microscopy plates containing minimal media with 0.08% glucose, and imaged over a period of 3 hours. In each of four replicate experiments, cells were imaged either every minute, every 90 seconds, or every 2 minutes. We used short exposure times (afforded by the highly abundant HTB2-GFP) and took only a single image per well per time point to reduce phototoxicity. We processed images with CalMorph, then matched cells across time

points by their centroid locations in the imaging fields. Overall, we obtained time series for 78 cells that each (1) were longer than 20 time points, (2) contained no gaps where the cell was not phenotyped for many consecutive time points, and (3) contained no images that appeared to be very out of focus, potentially resulting in misestimation of phenotype values. Because CalMorph divides cells into unbudded, small-budded, and large-budded stages, these 78 time series are also divided this way (11, 23, and 44 cells, respectively).

We used the Wishbone algorithm implemented in python [74] to estimate progression through the cell-division cycle. Wishbone recapitulates each of these 78 time series (S5 Fig) with Spearman correlations between the actual and inferred image orders that average 0.42, 0.85, and 0.40 across all unbudded, small-budded, or large-budded series, respectively. The lower correlations between Wishbone's predicted progress through division and time for the unbudded and large-budded cells may result because each time series captured only a part of the cell-division cycle and, during some stretches in the cycle, there are fewer morphological changes taking place. To estimate Wishbone's accuracy across a longer stretch of time, we merged the Wishbone predictions within the classes of unbudded, small-budded, or large-budded cell time series. To do so, we had to contend with the fact that the first time point for each imaged cell often represents a different moment in division. For example, some time series for unbudded cells start from an image that is already far along the division process (S5 Fig; values close to 1 on the vertical axis) whereas others start from a cell image that has just begun its division cycle (S5 Fig; values close to 0 on the vertical axis). Therefore, we aligned the time series by subtracting from each the difference between Wishbone's estimate of the average percent progress through division and the average time elapsed.

Note that, because this merging procedure utilized information from Wishbone, it imposes a correlation between time and Wishbone's estimated progress through division. To reduce the impact of this induced correlation, we eliminated the cell images in the middle of each time series, which represent the images that are most affected by this induced correlation. Eliminating 25% or 50% of cell images in this way reduced the correlations by at most 0.05, suggesting these correlations are not driven by our merging procedure.

## Assigning cells to a bin based on progression through cell division

We used Wishbone to estimate how far each fixed-cell image had progressed through cell division. Wishbone software requires input about which "start" cell has features resembling those present at the start of the cell cycle. To identify such features, we used the data from the live-imaged cell time series. We plotted how single-cell features change over the course of live imaging and chose several features that correlate best with progress through cell division (e.g., cell size, bud size, location of the nucleus). Complete datasets provided at Open Science Framework (DOI: 10.17605/OSF.IO/B7NY5) include information on which fixed-cell image was chosen as the start cell.

Using Wishbone's estimation of how far each fixed cell had progressed through division, we assigned each cell to one of 16 equal-sized bins. We did this separately for each of the 374 yeast strains, then merged like bins across strains, such that genetic diversity was constant across each of the final 16 bins. We obtained very similar results to those reported in Figs 4C, 4D and 5C when we used eight instead of 16 bins. The names of the traits plotted in Fig 4 represent succinct summaries of single-cell morphologies quantified using CalMorph [53]. For fuller descriptions of these traits, see the following trait designations in the CalMorph software manual: Fig 4A upper left: C11.1 in unbudded cells, C101 in budded cells; Fig 4A lower left: D184 in small-budded cells, D182 in unbudded and large-budded cells; Fig 4A upper right: C12.2; Fig 4A lower right: D116; Fig 4D left to right: C101 and C109 in small-budded cells,

C11.2 and D132 in small-budded cells, C105 and C113 in small-budded cells, C114 and D145 in large-budded cells, C109 and C126 in large-budded cells, D14.2 and D169 in large-budded cells.

## Eliminating genetic variation at the marker nearest a QTL

For each of the 27 QTL suspected of horizontal pleiotropy (i.e., pleiotropic QTL that influence at least one pair of traits for which $r_B$ significantly exceeds $r_W$), we divided the 374 phenotyped yeast strains into two groups based on whether they inherited the wine or the oak parent's allele at the genotyped marker closest to the QTL. In some cases, a QTL spans multiple markers; for example, a QTL on chromosome 15 that influences 64 morphological features spans 14 cM and four markers (S1 Table). These 64 genotype-phenotype associations are mainly clustered around the ninth marker on chromosome 15, though a few are closer to the eighth, 10th, or 11th. To avoid redundancy, for QTL spanning multiple markers, we study the one that is most represented. After dividing strains into two groups based on which allele they inherited at that marker, we performed correlation partitioning separately for each group of strains.

The names of the traits plotted in Fig 5B represent succinct summaries of single-cell morphologies quantified using CalMorph. For fuller descriptions of these traits, see the following trait designations in the CalMorph software manual: lower: D128 and C114 in large-budded cells; upper: D197 and D17.1 in large-budded cells.

## Quantifying trait correlations within each MA line

MA occurred in a diploid laboratory yeast strain with genotype *ade2*, *lys2-801*, *his3-ΔD200*, *leu2–3.112*, *ura3–52* [55]. Resulting diploid MA lines were sporulated to create haploids, which were sequenced in a previous study [56]. We previously imaged these haploid lines in high throughput (>1,000 clonal cells imaged per each of 94 lines) [51]. Fewer morphological traits were analyzed in that study than in the current study, such that there were only 3,731 pairs of traits to survey, as opposed to 5,645 in the QTL-mapping family. We calculated Pearson correlations between every pair of traits, separately within each MA line. The names of the traits plotted in Fig 6A represent succinct summaries of single-cell morphologies quantified using CalMorph. For fuller descriptions of these traits, see the following trait designations in the CalMorph software manual: upper left: D185 and D186 in large-budded cells; upper right: C102 and D132 in small-budded cells; lower left: C108 and D167 in large-budded cells; lower right: D135 and D169 in large-budded cells.

## Quantifying trait correlations across drug concentrations

For this analysis, we imaged the single-cell morphologies of 78 of the 374 strains that composed our QTL-mapping family. We chose these strains because they were stored together on a single 96-well plate (the rest of the 96 wells represent blanks or internal controls), removing concerns about batch effects. We imaged these strains after exponential growth in three concentrations of GdA (8.5 μM, 25 μM, and 100 μM). We chose these concentrations because of their wide-ranging impacts on cell growth rate [51]. We obtained single-cell morphology measurements for cells grown in the lowest concentrations of GdA from our previous study [51] and collected data for cells grown in higher concentrations following the procedures outlined in that study, which was very similar to those outlined above, but with a control for the solvent in which GdA is dissolved. Specifically, cells exposed to GdA were compared to cells imaged in identical conditions (containing the same concentration of the solvent DMSO) but lacking GdA. GdA+/−paired experiments are performed side by side, with cells grown in each condition being imaged in adjacent wells on a 384-well microscopy plate.

Resulting morphological data were analyzed following similar procedures as described above. Briefly, each trait was transformed via a Box-Cox transformation of the raw data based on the residuals of a linear model with strain, environment, and replicate as effects. Two replicates were performed for both the 8.5 and 100 μM environments and a single replicate for the 25 μM environment. Internal controls (wells representing the wine and oak parents) were used to correct for effects on phenotypic variation that resulted from differences among replicate experiments. We used WABA II to calculate cell-level ($r_W$) correlation coefficients in each of the three GdA concentrations, as well as the corresponding three control conditions. To calculate the impact of GdA on $r_W$, we compared $r_W$ in each drug versus control condition.

## Supporting information

**S1 Fig. Morphological differences exist between the parents of the QTL-mapping family.** (A–C) Each density plot displays the distribution of phenotype values from yeast cells corresponding to the wine parent (red), the oak parent (blue), or all of the 374 progeny (gray) for the trait listed on the horizontal axis. Trait names in parentheses correspond to those listed in the CalMorph manual [53]. Each distribution represents at minimum 5,000 cells from three replicate experiments; distributions corresponding to progeny strains represent many more cells (70,000–200,000 depending on whether the trait was measured in unbudded, small-budded, or large-budded cells). (D) The broad-sense heritability for each of the 155 morphological features for which QTL were detected. Heritability is low because cell morphology varies across the cell cycle, and so the amount of nongenetic phenotypic variation is high. Data underlying this figure can be found at https://osf.io/b7ny5/. QTL, quantitative trait loci. (TIFF)

**S2 Fig. Total numbers of cells imaged per each of 374 progeny strains.** Each point represents, for one of the 374 progeny strains, the number of unbudded, small-budded, or large-budded cells for which images passed filtering. Each box plot shows the median (center line), IQR (upper and lower hinges), and highest value within 1.5 × IQR (whiskers). Data underlying this figure can be found at https://osf.io/b7ny5/. IQR, interquartile range. (TIFF)

**S3 Fig. Comparison of correlation estimates obtained from correlation partitioning with those obtained from a mixed-effect linear model.** Each point represents one of 350 randomly sampled trait pairs of the 5,645 total. Vertical axes display trait correlations estimated using the correlation-partitioning approach; horizontal axes display trait correlations estimated using a mixed-effect linear model that specifies the variance-covariance structure of the experimental design. Data underlying this figure can be found at https://osf.io/b7ny5/. (TIFF)

**S4 Fig. Single-cell morphological traits have higher weighted clustering coefficients (*wcc*) than expected given the distribution of $r_W$.** (A–B) Force-directed networks visualizing how pairs of morphological features correlate across clones in unbudded (A) and small-budded (B) cells. Each node represents a single-cell morphological trait. The thickness of the line connecting each pair of nodes is proportional to $r_W$. Node position in the network is determined using the Fruchterman-Reingold algorithm. Purple nodes correspond to traits influenced by a QTL on chromosome 13 containing the *HOF1* gene. (C–D) Cumulative distributions of weighted clustering coefficients (*wcc*) in a network created using measured values of $r_W$ (red line) or in 100 permuted networks (gray lines) for traits corresponding to unbudded (C) or small-budded (D) cells. Permutations were performed by sampling $r_W$, without replacement, and reassigning each value to a random pair of traits. Data underlying this figure can be found at https://osf.io/

b7ny5/. QTL, quantitative trait loci.
(TIFF)

**S5 Fig. Wishbone recapitulates time-series data obtained in live images of 78 cells undergoing exponential growth.** Each point represents a cell image. Horizontal axes display the minute that image was captured during a 3-hour window of exponential growth. Vertical axes display Wishbone's prediction of how far that cell image has passed through the cell cycle. Linear regression lines are calculated with the "lm" method in the R package ggplot2 [99] and are colored red for images corresponding to unbudded cells, blue for small-budded cells, and purple for large-budded cells. Plots are organized by cell type and then from earliest to latest average predicted progress through cell division. Data underlying this figure can be found at https://osf.io/b7ny5/.
(TIFF)

**S6 Fig. Total numbers of cells imaged per strain in varying concentrations of GdA.** Each point represents, for one of the strains, the number of unbudded, small-budded, or large-budded cells for which images passed filtering. Each box plot shows the median (center line), IQR (upper and lower hinges), and highest value within $1.5 \times$ IQR (whiskers). Data underlying this figure can be found at https://osf.io/b7ny5/. GdA, geldanamycin; IQR, interquartile range.
(TIFF)

**S1 Table. Chromosomal locations, effects sizes, and phenotypes affected by quantitative trait loci described in this study.**
(CSV)

**S2 Table. Impact of gene swaps on single-cell morphological traits, including the corrected phenotypic difference between strains for each phenotype, and its standard deviation and standard error across replicate experiments.**
(CSV)

## Acknowledgments

We are grateful to Dmitri Petrov, Grant Kinsler, and Michael Lynch for helpful discussions. We also thank Barak Cohen and David Hall for providing strains used in this study.

## Author Contributions

**Conceptualization:** Kerry A. Geiler-Samerotte, Charalampos Lazaris, Annalise B. Paaby, Mark L. Siegal.

**Formal analysis:** Kerry A. Geiler-Samerotte, Charalampos Lazaris, Naomi Ziv, Mark L. Siegal.

**Funding acquisition:** Kerry A. Geiler-Samerotte, Mark L. Siegal.

**Investigation:** Kerry A. Geiler-Samerotte, Shuang Li, Austin Taylor, Chelsea Ramjeawan.

**Writing – original draft:** Kerry A. Geiler-Samerotte, Mark L. Siegal.

**Writing – review & editing:** Kerry A. Geiler-Samerotte, Shuang Li, Charalampos Lazaris, Austin Taylor, Naomi Ziv, Chelsea Ramjeawan, Annalise B. Paaby, Mark L. Siegal.

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
