## [Editor Report · Decision Letter 0]

30 Sep 2019

Dear Dr Geiler-Samerotte, 

Thank you for submitting your manuscript entitled "Extent and context dependence of pleiotropy revealed by high-throughput single-cell phenotyping" for consideration as a Research Article by PLOS Biology.

Your manuscript has now been evaluated by the PLOS Biology editorial staff, as well as by an academic editor with relevant expertise, and I'm writing to let you know that we would like to send your submission out for external peer review.

IMPORTANT: The Academic Editor has suggested that while submitted as a Research Article, this study may be better considered as a "Methods and Resources" paper, as this would enable reviewers to focus on the approach and substantial body of data, rather than the biological conclusions, which may be seen as somewhat nebulous. If you agree, please change the article type to "Methods and Resources."

Please re-submit your manuscript within two working days, i.e. by Oct 02 2019 11:59PM.

Kind regards,

Roli Roberts

Senior Editor

PLOS Biology

---

## [Decision Letter · Decision Letter 1]

4 Nov 2019

Dear Dr Geiler-Samerotte,

Thank you very much for submitting your manuscript "Extent and context dependence of pleiotropy revealed by high-throughput single-cell phenotyping" for consideration as a Methods and Resources at PLOS Biology. Your manuscript has been evaluated by the PLOS Biology editors, an Academic Editor with relevant expertise, and by three independent reviewers.

You'll see that while all three reviewers were impressed with the scale of your work, and reviewers #2 and #3 are broadly positive about your approach and the study, reviewer #1 raises some substantial conceptual objections. I asked the Academic Editor if s/he could provide any guidance as to how to approach these.

"As you are probably [aware], there are many ongoing conceptual debates related to defining pleiotropy. I think the authors raised an important point with the distinction of horizontal and vertical pleiotropy. I also like the way they approached this problem. ...rejecting the paper solely based on a semantic debate would not be the right decision. I do [note], however, that reviewer #1 raised several important issues. Most notably it would be important to study trait covariation across environments, at least on a few selected pairs of traits (ideally on isogenic cells)."

I hope this guidance is helpful; you should also address the concerns raised by the other two reviewers.

In light of the reviews (below), we will not be able to accept the current version of the manuscript, but we would welcome resubmission of a much-revised version that takes into account the reviewers' comments. We cannot make any decision about publication until we have seen the revised manuscript and your response to the reviewers' comments. Your revised manuscript is also likely to be sent for further evaluation by the reviewers.

Your revisions should address the specific points made by each reviewer. Please submit a file detailing your responses to the editorial requests and a point-by-point response to all of the reviewers' comments that indicates the changes you have made to the manuscript. In addition to a clean copy of the manuscript, please upload a 'track-changes' version of your manuscript that specifies the edits made. This should be uploaded as a "Related" file type. You should also cite any additional relevant literature that has been published since the original submission and mention any additional citations in your response. 

Before you revise your manuscript, please review the following PLOS policy and formatting requirements checklist PDF: http://journals.plos.org/plosbiology/s/file?id=9411/plos-biology-formatting-checklist.pdf. It is helpful if you format your revision according to our requirements - should your paper subsequently be accepted, this will save time at the acceptance stage.

Please note that as a condition of publication PLOS' data policy (http://journals.plos.org/plosbiology/s/data-availability) requires that you make available all data used to draw the conclusions arrived at in your manuscript. If you have not already done so, you must include any data used in your manuscript either in appropriate repositories, within the body of the manuscript, or as supporting information (N.B. this includes any numerical values that were used to generate graphs, histograms etc.). For an example see here: http://www.plosbiology.org/article/info%3Adoi%2F10.1371%2Fjournal.pbio.1001908#s5.

For manuscripts submitted on or after 1st July 2019, we require the original, uncropped and minimally adjusted images supporting all blot and gel results reported in an article's figures or Supporting Information files. We will require these files before a manuscript can be accepted so please prepare them now, if you have not already uploaded them. Please carefully read our guidelines for how to prepare and upload this data: https://journals.plos.org/plosbiology/s/figures#loc-blot-and-gel-reporting-requirements.

Upon resubmission, the editors will assess your revision and if the editors and Academic Editor feel that the revised manuscript remains appropriate for the journal, we will send the manuscript for re-review. We aim to consult the same Academic Editor and reviewers for revised manuscripts but may consult others if needed.

We expect to receive your revised manuscript within two months. Please email us (plosbiology@plos.org) to discuss this if you have any questions or concerns, or would like to request an extension. At this stage, your manuscript remains formally under active consideration at our journal; please notify us by email if you do not wish to submit a revision and instead wish to pursue publication elsewhere, so that we may end consideration of the manuscript at PLOS Biology.

When you are ready to submit a revised version of your manuscript, please go to https://www.editorialmanager.com/pbiology/ and log in as an Author. Click the link labelled 'Submissions Needing Revision' where you will find your submission record. 

Sincerely,

Roli Roberts

Senior Editor

PLOS Biology

REVIEWERS' COMMENTS:

Reviewer #1:

Pleiotropy refers to the common phenomenon that one mutation influences multiple phenotypic traits. In this work, Geiler-Samerotte and colleagues used yeast as a model system to study the biological basis of pleiotropy. They attempted to distinguish between so-called vertical and horizontal pleiotropy and reported detecting both types. They also used mutation accumulation lines to examine whether trait relationships are easily altered by random mutations and found the answer to be yes. They measured 167 yeast cell morphology traits in hundreds of thousands of cells by a high-throughput method, conducted QTL mapping of these traits in 374 segregants from a yeast cross, performed allele replacement experiments for a few genes, and carried out many computational analyses. The amount of work is impressive. However, I have major concerns about the definitions of vertical and horizontal pleiotropy and the method to separate them. In my view, these problems are so fundamental that their results are hard to interpret if one does not first resolve these conceptual issues. My detailed comments follow.

Major comments:

1. Pleiotropy is a characteristic of mutations, not traits. Thus, one must invoke mutations when defining pleiotropy. Yet, vertical pleiotropy was defined and studied in this work without mutations. Specifically, authors stated that “The key principle is that the distinction between vertical and horizontal pleiotropy lies in whether traits are correlated in the absence of genetic variation. For vertical pleiotropy, the answer is yes: because one trait influences the other or the two share an influence, non-genetic perturbations that alter one phenotype are expected to alter the other.” Here, they are defining and identifying vertical pleiotropy by examining whether two traits are inherently correlated in the absence of mutation. In their words and practice, pleiotropy is now a characteristic of pairs of traits, instead of mutations. They might argue that they are still studying mutations that simultaneously influence two traits. But if one already decides that two traits are inherently correlated (without even observing a mutation that affect them), what is the relevance of talking about the mutations affecting them? By their definition, such mutations will always be regarded as having vertical pleiotropy. I shall show below that this definition creates inconsistencies. Because horizontal pleiotropy is simply all pleiotropy that is not vertical, the definition of horizontal pleiotropy is also problematic. 

Let us consider the following example. A is the arm length of a human, B is the upper arm length, and C is the lower arm length. So, A=B+C. Clearly, A and B are inherently correlated, because increasing B will increase A. By contrast, B and C need not be inherently correlated. Now imagine a mutation that simultaneously increases B and C. So, it increases both B and A. Authors will claim that the mutation is vertically pleiotropic on A and B, because A and B are inherently correlated. But in fact, the mutation affects A not entirely through affecting B. We can imagine another mutation that increases B but decreases C by the same amount such that A is unaffected. Authors would still call this mutation vertically pleiotropic on A and B, because A and B are inherently correlated. But the mutation does not even affect A. This example illustrates the point that pleiotropy has to be defined for mutations rather than pairs of traits without considering specific mutations. 

The tomato example in the Introduction can make a similar point. The observation that chloroplasts affect coloration and sugar accumulation does that mean that a mutation that affects coloration and sugar accumulation must affect chloroplasts first. In other words, it is possible to have some mutations with vertical pleiotropy and some other mutations with horizontal pleiotropy on the two traits of coloration and sugar accumulation. 

Their definitions of vertical and horizontal pleiotropy and the method for separating them are thus problematic. With this fundamental conceptual flaw, the entire manuscript collapses. 

2. Even under the authors’ definitions of vertical and horizontal pleiotropy, their method of detecting vertical and horizontal pleiotropy is problematic. Specifically, they examine the covariation between two traits among isogenic cells only in the same environment. It is puzzling why they do not examine trait covariation across several environments. This would definitely increase the power of detecting covariation.

Related to the above point, they did a permutation test in the paragraph starting line 491 in order to see if rW is underestimated. They concluded that it is not, but this is a misinterpretation. The result only means that when there is no difference between groups, rB is not bigger than rW. It does not mean that rW is not underestimated.

3. I suggest that authors reanalyze their data using principal component traits, which are orthogonal to one another so are by definition not correlated. If a mutation affects two or more such traits, it is genuinely (or horizontally) pleiotropic. A critical question is what type of data one should use to define principal component traits. One choice would be morphologies of isogenic cells under the same environment. Another choice, which I think may be more reasonable, would be morphologies of isogenic cells under different environments. 

4. Their QTL mapping has only 225 markers and 374 segregants. So, the resolution is poor. Given that, one cannot claim to find a pleiotropic QTL simply from the finding that two traits are mapped to the same marker. The marker is unlikely to be causal for either trait; it is highly likely that the causal variants are different for the two traits. Authors are aware of this caveat so performed an allele replacement experiment in an attempt to prove that the QTL is pleiotropic. But each replaced segment of DNA contains the coding sequence of one gene plus 1 kb flanking sequence, amounting to about 2 kb in total. This sequence would have about 12 SNPs given the divergence of the two strains (0.006). In other words, even when they find that the replaced allele can explain morphological differences between the two strains, there is no guarantee that there is a pleiotropic mutation here (they correctly pointed out in the beginning of Introduction that pleiotropy is a feature of mutations, not genes). Furthermore, the allele replacement was done for only two genes, while 37 QTLs were found “pleiotropic”. While I understand the difficulty to find the causal genes and mutations of a phenotypic variation, the lack of verification of true pleiotropy of QTLs reported here means that all subsequent analyses are compromised, because they all depend on the unproven and most likely wrong assumption that the QTLs are pleiotropic. 

Minor comments:

5. Authors should mention vertical and horizontal pleiotropy in the title if this will remain as the focal point of the manuscript, because the current title is too general/vague.

6. Authors mentioned many times that they studied “thousands of trait pairs”. But these thousands of pairs are not independent from one another. There are probably <100 independent pairs here. So, it is better not to emphasize “thousands of trait pairs”.

7. The last sentence in the first paragraph of Introduction should be removed, because it is not about pleiotropy. Rather, it is about the number of genes affecting a trait. Having this sentence here would only create confusion. 

8. Line 149: “If there is modularity…” Authors imply that, without modularity, there cannot be horizontal pleiotropy. This is not true, as the above examples of arm length and tomato show. For this reason, the sentence on line 208 (“Collectively, the results…”) is also questionable. 

9. Line 226, it would be good to provide the strain names here.

10. The paragraph starting line 283. How did the authors predict the causal genes?

11. It is unclear whether each cell was measured for multiple traits. I got the impression that each cell was measured for all traits belonging to one of the three groups. Is this understanding right? If each cell is measured for only one trait, how did the authors compute correlations?

12. Authors used rW=0.2 as the cutoff to determine whether two traits are inherently correlated. What is the justification of using this cutoff? Cannot they use statistical significance in rW (FDR<0.05) as the cutoff? How would their conclusion change when the cutoff change?

13. They mentioned “correlation across clones” a number of time. Is this the same as correlation across cells of the same genotype? Please clarify.

14. Line 398. “In sum, pairs of traits with stronger correlations across clones (higher rW) are disproportionately represented among those influenced by pleiotropic QTL...” This is circular. How could this statement not be true? 

15. Line 502. “Contrary to this prediction, every morphological trait we surveyed varies MORE within strains than between strains.” Is MORE a typo? How could this be true?

16. Line 779. It is a pure coincidence that the median numbers of traits affected by a QTL are the same in mouse and yeast traits examined. These are very different traits and different organisms. Previous comparisons show no such equality between different types of traits even for the same species (e.g., Fig. 1 in ref. 11 cited).

Reviewer #2:

Geiler-Samerotte et al. addressed a fundamental and very important question on the nature of pleiotropy. Using yeast high-throughput single-cell imaging and quantitative genetics, the authors were able to decouple 'vertical' pleiotropy, caused by inherent constraints on phenotypes that impose correlations between traits, from 'horizontal' pleiotropy, where the action of a mutation or gene affects several traits that are otherwise not necessarily correlated. The approach is based on computing pairwise trait correlations both between (rb) and within (rw) genotypes.

The amount and quality of the experimental work is impressive, and the rigour and thoroughness of the analysis is even more spectacular. The authors applied fine methods to exclude possible artifacts (eg variation of trait-trait coupling during cell cycle) and they convincingly reach important conclusions: both types of pleiotropy exist, they can be decoupled by observing whether rb exceeds rw, and mutations can change trait-trait correlations. This latter conclusion is also reached by observing mutation-accumulation lines where rare cases of trait-trait correlations appeared in the course of neutral evolution. This is a very remarkable observation.

I have two requests of revision:

1) All along my reading, I was preoccupied by the possibility that rb could exceed rw due to reduced phenotypic variation within strains. The authors write that this is not the case (line 503) but they should provide a figure supporting this.

2) In the QTL mapping, a very large number of traits were tested individually for linkage. Text and methods do not clearly state whether the FDR 5% accounts for this multiplicity (it does account for genome-wide multiplicity but phenome-wide, does it?). The permutation test should be better explained.

Suggestions: line 131, change 'fair' by 'appropriate'; line 260, change 'individuals' by 'cells'.

Congratulations and thank you for this nice, well-written and helpful contribution to the field of genetics and evolution!

Reviewer #3:

This paper use a yeast-based model to investigate the extent of pleiotropy influenced by related traits (vertical) and by genetic differences (horizontal) . Using a single cell assay they produce an impressive dataset that enable the authors to explore interesting questions related to pleiotropy. 

MAJOR COMMENTS

- To further explore the pleiotropy concept, an unrelated phenotype should have been analysed (e.g. growth, drug resistance). The same cohort of segregants has been phenotyped for sporulation and used for QTL mapping. I’m surprised the authors did not look at pleiotropy in this (probably) unrelated phenotype vs. the one they measured and performed a comparison with the QTLs detected there (e.g. different QTLs are expected if no pleiotropy observed).

- The MA Lines (MALs) results are surprising. If I understood correctly, the MALs are diploids, so if most mutations are heterozygous, how can they affect so much the phenotypes (text lines 737-747). These extreme cases should be look more in details, e.g. what mutations are in these MALs? Can we exclude other events that occur at much higher rates (e.g. aneuplodies) can explain this? 

MINOR COMMENTS

- The collection of segregants has a gene, sps2 tagged with GFP, does this interfere with the phenotype measured here? Exact genotype of the strains should be reported to clarify this aspect.

- The collection of MALs should be briefly described (e.g. genotype, genetic background) etc. I guess is derived from lab strains, which is not an ideal comparison with the collection of segregants.

- Add grey scale to figure 1a, to indicate effect size, what about chromosome I and III? These should be plotted. Would this figure be more clear as a heat map (markers x traits), with colours reflecting QTL effect size?

- Are there vertically pleiotropic traits that are background dependent? If yes, can QTL be mapped

- The sample size is not super big, how this could affect the conclusion of the # of pleiotropic QTLs? Do you see a correlation with their effect sizes (e.g. stronger QTLs are more pleiotropic?).

---

## [Decision Letter · Decision Letter 2]

3 Jul 2020

Dear Dr Geiler-Samerotte,

Thank you for submitting your revised Methods and Resources entitled "Extent and context dependence of pleiotropy revealed by high-throughput single-cell phenotyping" for publication in PLOS Biology. I have now obtained advice from two of the original reviewers and have discussed their comments with the Academic Editor. Please accept my apologies for the unusual amount of time taken.

As mentioned before, reviewer #1 was not able to re-review this manuscript, but the other reviewers and the AE have assessed your responses to them, and are satisfied. Based on the reviews, we will probably accept this manuscript for publication, assuming that you will modify the manuscript to address the remaining point raised by reviewer #2. Please also make sure to address the data and other policy-related requests noted at the end of this email.

We expect to receive your revised manuscript within two weeks. Your revisions should address the specific points made by each reviewer. In addition to the remaining revisions and before we will be able to formally accept your manuscript and consider it "in press", we also need to ensure that your article conforms to our guidelines. A member of our team will be in touch shortly with a set of requests. As we can't proceed until these requirements are met, your swift response will help prevent delays to publication.

*Copyediting*

*Published Peer Review History*

*Early Version*

*Submitting Your Revision*

Sincerely,

Roli Roberts

Senior Editor

PLOS Biology

DATA POLICY:

Regardless of the method selected, please ensure that you provide the individual numerical values that underlie the summary data displayed in the figure panels as they are essential for readers to assess your analysis and to reproduce it.

My understanding is that most or all of the data displayed in your main and supplementary Figs are plotted directly from the R files in your OSF deposition; can you clarify whether this is the case (if not, please supply them separately). Please also include the OSF DOI or URL in the relevant legends (e.g. "Data underlying this Figure can be found at https://osf.io/b7ny5/").

REVIEWERS' COMMENTS:

Reviewer #2:

I previously requested two revisions: 1) showing the data supporting the substantial within-strain phenotypic variation and 2) controlling the FDR across the multiple traits.

Regarding point 1), the authors added in Fig 1 the distribution of heritability values (H2). This, indeed, addresses the point but they should write in the main text why. I strongly suggest to add a sentence after line 512 explaining why this distribution explains that rw is not low because of reduced within-strain variation. Something like "Indeed, broad-sense heritability of the traits did not exceed 15% (Fig 1B), reflecting that within-strain phenotypic variation accounted for at least 85% of the total variation and therefore exceeded the genetic (inter-strain) variation". This would clarify the preceeding sentence which is likely to surprise some readers (see Reviewer #1 point 15).

The authors addressed point 2 satisfyingly by reporting a more stringent set of QTLs and showing that this second set does not compromise the conclusions of the paper.

Reviewer #3:

The authors did an excellent job to revise the paper, well done.

---

## [Editor Report · Decision Letter 3]

31 Jul 2020

Dear Dr Geiler-Samerotte,

On behalf of my colleagues and the Academic Editor, Csaba Pál, I am pleased to inform you that we will be delighted to publish your Methods and Resources in PLOS Biology. 

Early Version

PRESS 

Kind regards,

Vita Usova

Publication Assistant, 

PLOS Biology

on behalf of

Roland Roberts,

Senior Editor

PLOS Biology